# Equivariant Flow Matching for Point Cloud Assembly

## Abstract

The goal of point cloud assembly is to reconstruct a complete 3D shape by aligning multiple point cloud pieces. This work presents a novel equivariant solver for assembly tasks based on flow matching models. We first theoretically show that the key to learning equivariant distributions via flow matching is to learn related vector fields. Based on this result, we propose an assembly model, called equivariant diffusion assembly (Eda), which learns related vector fields conditioned on the input pieces. We further construct an equivariant path for Eda, which guarantees high data efficiency of the training process. Our numerical results show that Eda is highly competitive on practical datasets, and it can even handle the challenging situation where the input pieces are non-overlapped.

## 1 Introduction

Point cloud (PC) assembly is a classic machine learning task which seeks to reconstruct 3D shapes by aligning multiple point cloud pieces. This task has been intensively studied for decades and has various applications such as scene reconstruction [48], robotic manipulation [32], cultural relics reassembly [39] and protein designing [41]. A key challenge in this task is to correctly align PC pieces with small or no overlap region, *i.e.*, when the correspondences between pieces are lacking.

To address this challenge, some recent methods [32, 40] utilized equivariance priors for pair-wise assembly tasks, *i.e.*, the assembly of two pieces. In contrast to most of the state-of-the-art methods [30, 51] which align PC pieces based on the inferred correspondence, these equivariant methods are correspondence-free, and they are guided by the equivariance law underlying the assembly task. As a result, these methods are able to assemble PCs without correspondence, and they enjoy high data efficiency and promising accuracy. However, the extension of these works to multi-piece assembly tasks remains largely unexplored.

In this work, we develop an equivariant method for multi-piece assembly based on flow matching [25]. Our main theoretical finding is that to learn an equivariant distribution via flow matching, one only needs to ensure that the initial noise is invariant and the vector field is related (Thm. 4.2). In other words, instead of directly handling the $SE(3)^N$-equivariance for $N$-piece assembly tasks, which can be computationally expensive, we only need to handle the related vector fields on $SE(3)^N$, which is efficient and easy to construct. Based on this result, we present a novel assembly model called equivariant diffusion assembly (Eda), which uses invariant noise and predicts related vector fields by construction. Eda is correspondence-free and is guaranteed to be equivariant by our theory. Furthermore, we construct a short and equivariant path for the training of Eda, which guarantees high data efficiency of the training process. When Eda is trained, an assembly solution can be sampled by numerical integration, *e.g.*, the Runge-Kutta method, starting from a random noise.

The contributions of this work are summarized as follows:

36 - We present an equivariant flow matching framework for multi-piece assembly tasks. Our theory
37   reduces the task of constructing equivariant conditional distributions to the task of constructing
38   related vector fields, thus it provides a feasible way to define equivariant flow matching models.

39 - Based on the theoretical result, we present a simple and efficient multi-piece PC assembly model,
40   called equivariant diffusion assembly (Eda), which is correspondence-free and is guaranteed to be
41   equivariant. We further construct an equivariant path for the training of Eda, which guarantees
42   high data efficiency.

43 - We numerically show that Eda produces highly accurate results on the challenging 3DMatch and
44   BB datasets, and it can even handle non-overlapped pieces.

## 2 Related work

46 Our proposed method is based on flow matching [25], which is one of the state-of-the-art diffusion
47 models for image generation tasks [11]. Some applications on manifolds have also been investi-
48 gated [4, 46]. Our model has two distinguishing features compared to the existing methods: it learns
49 conditional distributions instead of marginal distributions, and it explicitly incorporates equivariance
50 priors.

51 The PC assembly task studied in this work is related to various tasks in literature, such as PC
52 registration [30, 47], robotic manipulation [32, 31] and fragment reassembly [43]. All these tasks
53 aim to align the input PC pieces, but they are different in settings such as the number of pieces,
54 deterministic or probabilistic, and whether the PCs are overlapped. More details can be found in
55 Appx. A. In this work, we consider the most general setting: we aim to align multiple pieces of
56 non-overlapped PCs in a probabilistic way.

57 Recently, diffusion-based methods have been proposed for assembly tasks, such as registration [6,
58 18, 44] manipulation [32] and reassembly [34, 45]. However, most of these works simply regard the
59 solution space as a Euclidean space, where the underlying manifold structure and the equivariance
60 priors of the task are ignored. One notable exception is [32], which developed an equivariant diffusion
61 method for robotic manipulation, *i.e.*, pair-wise assembly tasks. Compared to [32], our method
62 is conceptually simpler because it does not require Brownian diffusion on $SO(3)$ whose kernel is
63 computationally intractable, and it solves the more general multi-piece problem. On the other hand,
64 the invariant flow theory has been studied in [20], which can be regarded as a special case of our
65 theory as discussed in Appx. C.1.

66 Another branch of related work is equivariant neural networks. Due to their ability to incorporate
67 geometric priors, this type of networks has been widely used for processing 3D graph data such
68 as PCs and molecules. In particular, E3NN [14] is a well-known equivariant network based on the
69 tensor product of the input and the edge feature. An acceleration technique for E3NN was recently
70 proposed [28]. On the other hand, the equivariant attention layer was studied in [12, 22, 24]. Our
71 work is related to this line of approach, because our diffusion network can be seen as an equivariant
72 network with an additional time parameter.

## 3 Preliminaries

74 This section introduces the major tools used in this work. We first define the equivariances in Sec. 3.1,
75 then we briefly recall the flow matching model in Sec. 3.2.

### 3.1 Equivariances of PC assembly

77 Consider the action $G = \prod_{i=1}^{N} SE(3)$ on a set of $N$ ($N \geq 2$) PCs $X = \{X_1, \ldots, X_N\}$, where
78 $SE(3)$ is the 3D rigid transformation group, $\prod$ is the direct product, and $X_i$ is the i-th PC piece
79 in 3D space. We define the action of $\boldsymbol{g} = (g_1, \ldots, g_N) \in G$ on $X$ as $\boldsymbol{g}X = \{g_i X_i\}_{i=1}^{N}$, *i.e.*, each
80 PC $X_i$ is rigidly transformed by the corresponding $g_i$. For the rotation subgroup $SO(3)^N$, the
81 action of $\boldsymbol{r} = (r_1, \ldots, r_N) \in SO(3)^N$ on $X$ is $\boldsymbol{r}X = \{r_i X_i\}_{i=1}^{N}$. For $SO(3) \subseteq G$, we denote
82 $r = (r, \ldots, r) \in SO(3)$ for simplicity, and the action of $r$ on $X$ is written as $rX = \{r X_i\}_{i=1}^{N}$.

83 We also consider the permutation of $X$. Let $S_N$ be the permutation group of $N$, the action of $\sigma \in S_N$
84 on $X$ is $\sigma X = \{X_{\sigma(i)}\}_{i=1}^{N}$, and the action on $\boldsymbol{g}$ is $\sigma \boldsymbol{g} = (g_{\sigma(1)}, \ldots, g_{\sigma(N)})$. For group multiplication,

we denote $\mathcal{R}_{(\cdot)}$ the right multiplication and $\mathcal{L}_{(\cdot)}$ the left multiplication, *i.e.*, $(\mathcal{R}_{\boldsymbol{r}})\boldsymbol{r}' = \boldsymbol{r}'\boldsymbol{r}$, and $(\mathcal{L}_{\boldsymbol{r}})\boldsymbol{r}' = \boldsymbol{r}\boldsymbol{r}'$ for $\boldsymbol{r}, \boldsymbol{r}' \in SO(3)^N$.

In our setting, for the given input $X$, the solution to the assembly task is a conditional distribution $P_X \in \mu(G)$, where $\mu(G)$ is the set of probability distribution on $G$. We study the following three equivariances of $P_X$ in this work:

**Definition 3.1.** Let $P_X \in \mu(G)$ be a probability distribution on $G = SE(3)^N$ conditioned on $X$, and let $(\cdot)_\#$ be the pushforward of measures.

- $P_X$ is $SO(3)^N$-equivariant if $(\mathcal{R}_{\boldsymbol{r}^{-1}})_\# P_X = P_{\boldsymbol{r}X}$ for $\boldsymbol{r} \in SO(3)^N$.

- $P_X$ is permutation-equivariant if $\sigma_\# P_X = P_{\sigma X}$ for $\sigma \in S_N$.

- $P_X$ is $SO(3)$-invariant if $(\mathcal{L}_r)_\# P_X = P_X$ for $r \in SO(3)$.

Intuitively, the equivariances defined in Def. 3.1 are three natural priors of the assembly task: the $SO(3)^N$-equivariance of $P_X$ implies that the solution will be properly transformed when $X$ is rotated; the permutation-equivariance of $P_X$ implies that the assembled shape is independent of the order of $X$; and the $SO(3)$-invariance of $P_X$ implies that the solution does not have a preferred orientation.

Note that when $N = 2$, $SO(3)^N$-equivariance is closely related to $SE(3)$-bi-equivariance [32, 40], and permutation-equivariance becomes swap-equivariance in [40]. Detailed explanations can be found in Appx. B.

We finally recall the definition of $SO(3)$-equivariant networks, which will be the main computational tool of this work. We call $F^l \in \mathbb{R}^{2l+1}$ a degree-$l$ $SO(3)$-equivariant feature if the action of $r \in SO(3)$ on $F^l$ is the matrix-vector production: $rF^l = R^l F^l$, where $R^l \in \mathbb{R}^{(2l+1)\times(2l+1)}$ is the degree-$l$ Wigner-D matrix of $r$. We call a network $w$ $SO(3)$-equivariant if it maintains the equivariance from the input to the output: $w(rX) = rw(X)$, where $w(X)$ is a $SO(3)$-equivariant feature. More detailed introduction of equivariances and the underlying representation theory can be found in [3].

## 3.2 Vector fields and flow matching

To sample from a data distribution $P_1 \in \mu(M)$, where $M$ is a smooth manifold (we only consider $M = G$ in this work), the flow matching [25] approach constructs a time-dependent diffeomorphism $\phi_\tau : M \to M$ satisfying $(\phi_0)_\# P_0 = P_0$ and $(\phi_1)_\# P_0 = P_1$, where $P_0 \in \mu(M)$ is a fixed noise distribution, and $\tau \in [0, 1]$ is the time parameter. Then the sample of $P_1$ can be represented as $\phi_1(g)$ where $g$ is sampled from $P_0$.

Formally, $\phi_\tau$ is defined as a flow, *i.e.*, an integral curve, generated by a time-dependent vector field $v_\tau : M \to TM$, where $TM$ is the tangent bundle of $M$:

$$
\begin{aligned}
\frac{\partial}{\partial \tau} \phi_\tau(\boldsymbol{g}) &= v_\tau(\phi_\tau(\boldsymbol{g})), \\
\phi_0(\boldsymbol{g}) &= \boldsymbol{g}, \quad \forall \boldsymbol{g} \in M.
\end{aligned}
\tag{1}
$$

According to [25], an efficient way to construct $v_\tau$ is to define a path $h_\tau$ connecting $P_0$ to $P_1$. Specifically, let $\boldsymbol{g}_0$ and $\boldsymbol{g}_1$ be samples from $P_0$ and $P_1$ respectively, and $h_0 = \boldsymbol{g}_0$ and $h_1 = \boldsymbol{g}_1$. $v_\tau$ can be constructed as the solution to the following problem:

$$
\min_v \mathbb{E}_{\tau, \boldsymbol{g}_0 \sim P_0, \boldsymbol{g}_1 \sim P_1} \| v_\tau(h_\tau) - \frac{\partial}{\partial \tau} h_\tau \|_F^2.
\tag{2}
$$

When $v$ is learned using (2), we can obtain a sample from $P_1$ by first sampling a noise $\boldsymbol{g}_0$ from $P_0$ and then taking the integral of (1).

In this work, we consider a family of vector fields, flows and paths conditioned on the given PC, and we use the pushforward operator on vector fields to study their relatedness [37]. Formally, let $F : M \to M$ be a diffeomorphism, $v$ and $w$ be vector fields on $M$. $w$ is $F$-related to $v$ if $w(F(\boldsymbol{g})) = F_{*,\boldsymbol{g}} v(\boldsymbol{g})$ for all $\boldsymbol{g} \in M$, where $F_{*,\boldsymbol{g}}$ is the differential of $F$ at $\boldsymbol{g}$. Note that we denote $v_X$, $\phi_X$ and $h_X$ the vector field, flow and path conditioned on PC $X$ respectively.

## 4  Method

In this section, we provide the details of the proposed Eda model. First, the PC assembly problem is formulated in Sec. 4.1. Then, we parametrize related vector fields in Sec. 4.2. The training and sampling procedures are finally described in Sec. 4.3 and Sec. 4.4 respectively.

### 4.1  Problem formulation

Given a set $X$ containing $N$ PC pieces, *i.e.*, $X = \{X_i\}_{i=1}^N$ where $X_i$ is the $i$-th piece, the goal of assembly is to learn a distribution $P_X \in \mu(G)$, *i.e.*, for any sample $\boldsymbol{g}$ of $P_X$, $\boldsymbol{g}X$ should be the aligned complete shape. We assume that $P_X$ has the following equivariances:

**Assumption 4.1.** $P_X$ is $SO(3)^N$-equivariant, permutation-equivariant and $SO(3)$-invariant.

We seek to approximate $P_X$ using flow matching. To avoid translation ambiguity, we also assume that, without loss of generality, the aligned PCs $\boldsymbol{g}X$ and each input piece $X_i$ are centered, *i.e.*, $\sum_i \mathbf{m}(g_i X_i) = 0$, and $\mathbf{m}(X_i) = 0$ for all $i$, where $\mathbf{m}(\cdot)$ is the mean vector.

### 4.2  Equivariant flow

The major challenge in our task is to ensure the equivariance of the learned distribution, because a direct implement of flow matching (1) generally does not guarantee any equivariance. To address this challenge, we utilize the following theorem, which claims that when the noise distribution $P_0$ is invariant and vector fields $v_X$ are related, the pushforward distribution $(\phi_X)\#P_0$ is guaranteed to be equivariant.

**Theorem 4.2.** *Let $G$ be a smooth manifold, $F : G \to G$ be a diffeomorphism, and $P \in \mu(G)$. If vector field $v_X \in TG$ is $F$-related to vector field $v_Y \in TG$, then*

$$F_\# P_X = P_Y, \tag{3}$$

*where $P_X = (\phi_X)_\# P_0$, $P_Y = (\phi_Y)_\#(F_\# P_0)$. Here $\phi_X, \phi_Y : G \to G$ are generated by $v_X$ and $v_Y$ respectively.*

Specifically, Thm. 4.2 provides a concrete way to construct equivariant distributions as follow.

**Assumption 4.3** (Invariant noise). $P_0$ is $SO(3)^N$-invariant, permutation-invariant and $SO(3)$-invariant, *i.e.*, $(\mathcal{R}_{\boldsymbol{r}^{-1}})_\# P_0 = P_0$, $\sigma_\# P_0 = P_0$ and $P_0 = (\mathcal{L}_r)_\# P_0$ for $\boldsymbol{r} \in SO(3)^N$, $\sigma \in S_N$ and $r \in SO(3)$.

**Corollary 4.4.** *Under assumption 4.3,*

- *if $v_X$ is $\mathcal{R}_{\boldsymbol{r}^{-1}}$-related to $v_{\boldsymbol{r}X}$, then $(\mathcal{R}_{\boldsymbol{r}^{-1}})_\# P_X = P_{\boldsymbol{r}X}$, where $P_X = (\phi_X)_\# P_0$ and $P_{\boldsymbol{r}X} = (\phi_{\boldsymbol{r}X})_\# P_0$. Here $\phi_X, \phi_{\boldsymbol{r}X} : G \to G$ are generated by $v_X$ and $v_{\boldsymbol{r}X}$ respectively.*

- *if $v_X$ is $\sigma$-related to $v_{\sigma X}$, then $\sigma_\# P_X = P_{\sigma X}$, where $P_X = (\phi_X)_\# P_0$ and $P_{\sigma X} = (\phi_{\sigma X})_\# P_0$. Here $\phi_X, \phi_{\sigma X} : G \to G$ are generated by $v_X$ and $v_{\sigma X}$ respectively.*

- *if $v_X$ is $\mathcal{L}_r$-invariant, i.e., $v_X$ is $\mathcal{L}_r$-related to $v_X$, then $(\mathcal{L}_r)_\# P_X = P_X$, where $P_X = (\phi_X)_\# P_0$.*

Now we construct the vector field required by Cor. 4.4. We start by constructing $(\mathcal{R}_{\boldsymbol{g}^{-1}})$-related vector fields, which are $(\mathcal{R}_{\boldsymbol{r}^{-1}})$-related by definition, where $\boldsymbol{g} \in SE(3)^N$ and $\boldsymbol{r} \in SO(3)^N$. Specifically, we have the following proposition:

**Proposition 4.5.** $v_X$ *is $\mathcal{R}_{\boldsymbol{g}^{-1}}$-related to $v_{\boldsymbol{g}X}$ if and only if $v_X(\boldsymbol{g}) = (\mathcal{R}_{\boldsymbol{g}})_{*,e} v_{\boldsymbol{g}X}(e)$ for all $\boldsymbol{g} \in SE(3)^N$.*

According to Prop. 4.5, to construct a $(\mathcal{R}_{\boldsymbol{g}^{-1}})$-related vector field $v_X$, we only need to parametrize $v_X$ at the identity $e$. Specifically, let $f$ be a neural network parametrizing $v_X(e)$, *i.e.*, $f(X) = v_X(e)$, we can define $v_X$ as

$$v_X(\boldsymbol{g}) = (\mathcal{R}_{\boldsymbol{g}})_{*,e} f(\boldsymbol{g}X). \tag{4}$$

Here, $f(X) \in \mathfrak{se}(3)^N$ takes the form of

$$f(X) = \bigoplus_{i=1}^N f_i(X) \quad \text{where} \quad f_i(X) = \begin{pmatrix} w_\times^i(X) & t^i(X) \\ 0 & 0 \end{pmatrix} \in \mathfrak{se}(3) \subseteq \mathbb{R}^{4 \times 4}. \tag{5}$$

The rotation component $w_\times^i(X) \in \mathbb{R}^{3\times3}$ is a skew matrix with elements in the vector $w^i(X) \in \mathbb{R}^3$, and $t^i(X) \in \mathbb{R}^3$ is the translation component. For simplicity, we omit the superscript $i$ when the context is clear.

Then we enforce the other two relatedness of $v_X$ (4). According to the following proposition, $\sigma$-relatedness can be guaranteed if $f$ is permutation-equivariant, and $\mathcal{L}_r$-invariance can be guaranteed if both $w$ and $t$ are SO(3)-equivariant.

**Proposition 4.6.** *For $v_X$ defined in (4),*

- *if $f$ is permutation-equivariant, i.e., $f(\sigma X) = \sigma f(X)$ for $\sigma \in S_N$ and PCs $X$, then $\sigma_\# v_X = v_{\sigma X}$;*

- *if $f$ is SO(3)-equivariant, i.e., $w(rX) = rw(X)$ and $t(rX) = rt(X)$ for $r \in SO(3)$ and PCs $X$, then $(\mathcal{L}_r)_\# v_X = v_{rX}$.*

Finally, we define $P_0 = (U_{SO(3)} \otimes \mathcal{N}(0, \omega I))^N$, where $U_{SO(3)}$ is the uniform distribution on $SO(3)$, $\mathcal{N}$ is the normal distribution on $\mathbb{R}^3$ with mean zero and isotropic variance $\omega \in \mathbb{R}_+$, and $\otimes$ represents the independent coupling. It is straightforward to verify that $P_0$ indeed satisfies assumption 4.3.

In summary, with $P_0$ defined above and $f$ (5) satisfying the assumptions in Prop. 4.6, Theorem 4.2 guarantees that the learned distribution has the desired equivariances, *i.e.*, $SO(3)^N$-equivariance, permutation-equivariance and $SO(3)$-invariance.

## 4.3 Training

To learn the vector field $v_X$ (4) using flow matching (2), we now need to define $h_X$, and the sampling strategy of $\tau$, $\boldsymbol{g}_0$ and $\boldsymbol{g}_1$. A canonical choice [4] is $h(\tau) = \boldsymbol{g}_0 \exp(\tau \log(\boldsymbol{g}_0^{-1} \boldsymbol{g}_1))$, where $\boldsymbol{g}_0$ and $\boldsymbol{g}_1$ are sampled independently, and $\tau$ is sampled from a predefined distribution, *e.g.*, the uniform distribution $U_{[0,1]}$. However, this definition of $h$, $\boldsymbol{g}_0$ and $\boldsymbol{g}_1$ does not utilize any equivariance property of $v_X$, thus it does not guarantee a high data efficiency.

To address this issue, we construct a "short" and equivariant $h_X$ in the following two steps. First, we independently sample $\boldsymbol{g}_0$ from $P_0$ and $\tilde{\boldsymbol{g}}_1$ from $P_X$, and obtain $\boldsymbol{g}_1 = r^* \tilde{\boldsymbol{g}}_1$, where $r^* \in SO(3)$ is a rotation correction of $\tilde{\boldsymbol{g}}_1$:

$$r^* = \arg\min_{r \in SO(3)} ||r\tilde{\boldsymbol{g}}_1 - \boldsymbol{g}_0||_F^2. \tag{6}$$

Then, we define $h_X$ as

$$h_X(\tau) = \exp(\tau \log(\boldsymbol{g}_1 \boldsymbol{g}_0^{-1}))\boldsymbol{g}_0. \tag{7}$$

We call $h_X$ (7) a path generated by $\boldsymbol{g}_0$ and $\tilde{\boldsymbol{g}}_1$. Note that $h_X$ (7) is a well-defined path connecting $\boldsymbol{g}_0$ to $\boldsymbol{g}_1$, because $h_X(0) = \boldsymbol{g}_0$ and $h_X(1) = \boldsymbol{g}_1$, and $\boldsymbol{g}_1$ follows $P_X$ (Prop. C.5).

The advantages of $h_X$ (7) are twofold. First, instead of connecting a noise $\boldsymbol{g}_0$ to an independent data sample $\tilde{\boldsymbol{g}}_1$, $h_X$ connects $\boldsymbol{g}_0$ to a modified sample $\boldsymbol{g}_1$ where the redundant rotation component is removed, thus it is easier to learn. Second, the velocity fields of $h_X$ enjoy the same relatedness as $v_X$ (4), which leads to high data efficiency. Formally, we have the following observation.

**Proposition 4.7** (Data efficiency). *Under assumption 4.3, 4.1, and C.4, we further assume that $v_X$ satisfies the relatedness property required in Cor. 4.4, i.e., $v_X$ is $\mathcal{R}_{\boldsymbol{r}^{-1}}$-related to $v_{\boldsymbol{r}X}$, $v_X$ is $\sigma$-related to $v_{\sigma X}$, and $v_X$ is $\mathcal{L}_r$-invariant. Denote $L(X) = \mathbb{E}_{\tau,\boldsymbol{g}_0 \sim P_0, \tilde{\boldsymbol{g}}_1 \sim P_X}||v_X(h_X(\tau)) - \frac{\partial}{\partial \tau}h_X(\tau)||_F^2$ the training loss (2) of PC $X$, where $h_X$ is generated by $\boldsymbol{g}_0$ and $\tilde{\boldsymbol{g}}_1$ as defined in (7). Then*

- *$L(X) = L(\boldsymbol{r}X)$ for $\boldsymbol{r} \in SO(3)^N$.*

- *$L(X) = L(\sigma X)$ for $\sigma \in S_N$.*

- *$L(X) = \hat{L}(X)$, where $\hat{L}(X) = \mathbb{E}_{\tau,\boldsymbol{g}_0' \sim P_0, \tilde{\boldsymbol{g}}_1' \sim (\mathcal{L}_r)_\# P_X}||v_X(h_X(\tau)) - \frac{\partial}{\partial \tau}h_X(\tau)||_F^2$ is the loss where the data distribution $P_X$ is pushed forward by $\mathcal{L}_r \in SO(3)$.*

Prop. 4.7 implies that when $h_X$ (7) is combined with the equivariant components developed in Sec. 4.2, the following three data augmentations are not needed: 1) random rotation of each input piece $X_i$, 2) random permutation of the order of the input pieces, and 3) random rotation of the assembled shape, because they have no influence on the training loss.

## 4.4 Sampling via the Runge-Kutta method

Finally, when the vector field $v_X$ (4) is learned, we can obtain a sample $\boldsymbol{g}_1$ from $P_X$ by numerically integrating $v_X$ starting from a noise $\boldsymbol{g}_0$ from $P_0$. In this work, we use the Runge-Kutta (RK) solver on $SE(3)^N$, which is a generalization of the classical RK solver on Euclidean spaces. For clarity, we present the formulations below, and refer the readers to [7] for more details.

To apply the RK method, we first discretize the time interval $[0, 1]$ into $I$ steps, $i.e.$, $\tau_i = \frac{i}{I}$ for $i = 0, \ldots, I$, with a step length $\eta = \frac{1}{I}$. For the given input $X$, denote $f(\boldsymbol{g}X)$ at time $\tau$ by $f_\tau(\boldsymbol{g})$ for simplicity. The first-order RK method (RK1), $i.e.$, the Euler method, is to iterate:

$$\boldsymbol{g}_{i+1} = \exp(\eta f_{\tau_i}(\boldsymbol{g}_i))\boldsymbol{g}_i, \tag{8}$$

for $i = 0, \ldots, I$. To achieve higher accuracy, we can use the fourth-order RK method (RK4):

$$k_1 = f_{\tau_i}(\boldsymbol{g}_i),\ k_2 = f_{\tau_i + \frac{1}{2}\eta}\big(\exp(\frac{1}{2}\eta k_1)\boldsymbol{g}_i\big),\ k_3 = f_{\tau_i + \frac{1}{2}\eta}\big(\exp(\frac{1}{2}\eta k_2)\boldsymbol{g}_i\big),\ k_4 = f_{\tau_i + \eta}\big(\exp(\eta k_3)\boldsymbol{g}_i\big),$$

$$\boldsymbol{g}_{i+1} = \exp(\frac{1}{6}\eta k_4)\exp(\frac{1}{3}\eta k_3)\exp(\frac{1}{3}\eta k_2)\exp(\frac{1}{6}\eta k_1)\boldsymbol{g}_i. \tag{9}$$

Note that RK4 (9) is more computationally expensive than RK1 (8), because it requires four evaluations of $v_X$ at different points at each step, $i.e.$, four forward passes of network $f$, while the Euler method only requires one evaluation per step.

# 5 Implementation

This section provides the details of the network $f$ (5). Our design principle is to imitate the standard transformer structure [38] to retain its best practices. In addition, according to Prop. 4.6, we also require $f$ to be permutation-equivariant and $SO(3)$-equivariant.

The overall structure of the proposed network is shown in Fig. 1. In a forward pass, the input PC pieces $\{X_i\}_{i=1}^N$ are first downsampled using a few downsampling blocks, and then fed into the Croco blocks [42] to model their relations. Meanwhile, the time step $\tau$ is first embedded using a multi-layer perceptron (MLP) and then incorporated into the above blocks via adaptive normalization [29]. The output is finally obtained by a piece-wise pooling.

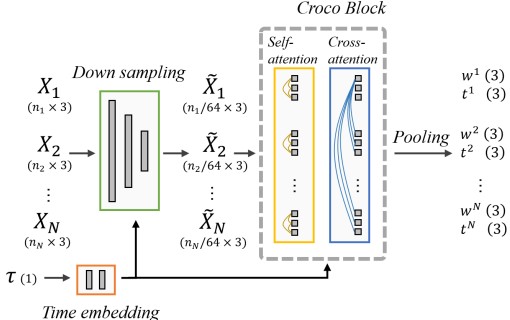

Figure 1: An overview of our model. The shapes of variables are shown in the brackets.

Next, we provide details of the equivariant attention layers, which are the major components of both the downsampling block and the Croco block, in Sec. 5.1. Other layers, including the nonlinear and normalization layers, are described in Sec. 5.2.

## 5.1 Equivariant attention layers

Let $F_u^l \in \mathbb{R}^{c \times (2l+1)}$ be a channel-$c$ degree-$l$ feature at point $u$. The equivariant dot-product attention is defined as:

$$A_u^l = \sum_{v \in KNN(u) \setminus \{u\}} \frac{\exp\left(\langle Q_u, K_{vu}\rangle\right)}{\sum_{v' \in KNN(u) \setminus \{u\}} \exp\left(\langle Q_u, K_{v'u}\rangle\right)} V_{vu}^l, \tag{10}$$

where $\langle \cdot, \cdot \rangle$ is the dot product, $KNN(u) \subseteq \bigcup_i X_i$ is a subset of points $u$ attends to, $K, V \in \mathbb{R}^{c \times (2l+1)}$ take the form of the e3nn [14] message passing, and $Q \in \mathbb{R}^{c \times (2l+1)}$ is obtained by a linear transform:

$$Q_u = \bigoplus_l W_Q^l F_u^l, \quad K_v = \bigoplus_l \sum_{l_e, l_f} c_K^{(l, l_e, l_f)}(|uv|) Y^{l_e}(\widehat{vu}) \otimes_{l_e, l_f}^l F_v^{l_f}, \tag{11}$$

$$V_v^l = \sum_{l_e, l_f} c_V^{(l, l_e, l_f)}(|uv|) Y^{l_e}(\widehat{vu}) \otimes_{l_e, l_f}^l F_v^{l_f}. \tag{12}$$

Here, $W_Q^l \in \mathbb{R}^{c \times c}$ is a learnable weight, $|vu|$ is the distance between point $v$ and $u$, $\widehat{vu} = \vec{vu}/|vu| \in \mathbb{R}^3$ is the normalized direction, $Y^l : \mathbb{R}^3 \rightarrow \mathbb{R}^{2l+1}$ is the degree-$l$ spherical harmonic function, $c : \mathbb{R}_+ \rightarrow \mathbb{R}$ is a learnable function that maps $|vu|$ to a coefficient, and $\otimes$ is the tensor product with the Clebsch-Gordan coefficients.

To accelerate the computation of $K$ and $V$, we use the $SO(2)$-reduction technique [28], which rotates the edge $uv$ to the $y$-axis, so that the computation of spherical harmonic function, the Clebsch-Gordan coefficients, and the iterations of $l_e$ are no longer needed. More details are provided in Appx. D.

Following Croco [42], we stack two types of attention layers, *i.e.*, the self-attention layer and the cross-attention layer, into a Croco block to learn the features of each PC piece while incorporating information from other pieces. For self-attention layers, we set *KNN(u)* to be the $k$-nearest neighbors of $u$ in the same piece, and for cross-attention layers, we set *KNN(u)* to be the $k$-nearest neighbors of $u$ in each of the different pieces. In addition, to reduce the computational cost, we use downsampling layers to reduce the number of points before the Croco layers. Each downsampling layer consists of a farthest point sampling (FPS) layer and a self-attention layer.

## 5.2 Adaptive normalization and nonlinear layers

Following the common practice [10], we seek to use the GELU activation function [16] in our transformer structure. However, GELU in its original form is not $SO(3)$-equivariant. To address this issue, we adopt a projection formulation similar to [9]. Specifically, we define the equivariant GELU (Elu) as:

$$Elu(F^l) = GELU(\langle F^l, \widehat{WF^l} \rangle) \tag{13}$$

where $\widehat{x} = x/\|x\|$ is the normalized feature, $W \in \mathbb{R}^{c \times c}$ is a learnable weight. Note that Elu (13) is a natural extension of GELU, because when $l = 0$, $Elu(F^0) = GELU(\pm F^0)$.

As for the normalization layers, we use RMS-type layer normalization layers [50] following [23], and we use the adaptive normalization [29] technique to incorporate the time step $\tau$. Specifically, we use the adaptive normalization layer *AN* defined as:

$$AN(F^l, \tau) = F^l/\sigma \cdot MLP(\tau), \tag{14}$$

where $\sigma = \sqrt{\frac{1}{c \cdot l_{max}} \sum_{l=1}^{l_{max}} \frac{1}{2l+1} \langle F^l, F^l \rangle}$, $l_{max}$ is the maximum degree, and *MLP* is a multi-layer perceptron that maps $\tau$ to a vector of length $c$.

We finally remark that the network $\boldsymbol{f}$ defined in this section is $SO(3)$-equivariant because each layer is $SO(3)$-equivariant by construction. $\boldsymbol{f}$ is also permutation-equivariant because it does not use any order information of $X_i$.

# 6 Experiment

This section evaluates Eda on practical assembly tasks. After introducing the experiment settings in Sec. 6.1, we first evaluate Eda on the pair-wise registration tasks in Sec. 6.2, and then we consider the multi-piece assembly tasks in Sec. 6.3. An ablation study on the number of PC pieces is finally presented in Sec. 6.4.

## 6.1 Experiment settings

We evaluate the accuracy of an assembly solution using the averaged pair-wise error. For a predicted assembly $\boldsymbol{g}$ and the ground truth $\hat{\boldsymbol{g}}$, the rotation error $\Delta r$ and the translation error $\Delta t$ are computed as: $(\Delta r, \Delta t) = \frac{1}{N(N-1)} \sum_{i \neq j} \tilde{\Delta}(\hat{g}_i, \hat{g}_j g_j^{-1} g_i)$, where the pair-wise error $\tilde{\Delta}$ is computed as $\tilde{\Delta}(g, \hat{g}) = \left( \frac{180}{\pi} accos \left( \frac{1}{2} \left( tr(r\hat{r}^T) - 1 \right) \right), \|\hat{t} - t\| \right)$. Here $g = (r, t)$, $\hat{g} = (\hat{r}, \hat{t})$, and $tr(\cdot)$ represents the trace.

For Eda, we use 2 Croco blocks, and 4 downsampling layers with a downsampling ratio $0.25$. We use $k = 10$ nearest neighbors, $l_{max} = 2$ degree features with $d = 64$ channels and 4 attention heads. Following [29], we keep an exponential moving average (EMA) with a decay of $0.99$, and we use the AdamW [26] optimizer with a learning rate $10^{-4}$. Following [11], we use a logit-normal sampling for time variable $\tau$. For each experiment, we train Eda on 3 Nvidia A100 GPUs for at most 5 days. We denote Eda with $q$ steps of RK$p$ as "Eda (RK$p$, $q$)" , *e.g.*, Eda (RK1, 10) represents Eda with 10 steps of RK1.

## 6.2 Pair-wise registration

This section evaluates Eda on rotated 3DMatch [48] (3DM) dataset containing PC pairs from indoor scenes. Following [17], we consider the 3DLoMatch split (3DL), which contains PC pairs with smaller overlap ratios.

Table 1: The overlap ratio of PC pairs (%).

|  | 3DM | 3DL | 3DZ |
|---|---|---|---|
| Training set | (10, 100) | | 0 |
| Test set | (30, 100) | (10, 30) | 0 |

Furthermore, to highlight the ability of Eda on non-overlapped assembly tasks, we consider a new split called 3DZeroMatch (3DZ), which contains non-overlapped PC pairs. The comparison of these three splits is shown in Tab. 1.

We compare Eda against the following baseline methods: FGR [52], GEO [30], ROI [47], and AMR [6], where FGR is a classic optimization-based method, GEO and ROI are correspondence-based methods, and AMR is a recently proposed diffusion-like method based on GEO. We report the results of the baseline methods using their official implementations. Note that the correspondence-free methods like [32, 40] do not scale to this dataset.

Table 2: Quantitative results on rotated 3DMatch. ROI (n): ROI with $n$ RANSAC samples.

|  | 3DM | | 3DL | | 3DZ | |
|---|---|---|---|---|---|---|
|  | $\Delta r$ | $\Delta t$ | $\Delta r$ | $\Delta t$ | $\Delta r$ | $\Delta t$ |
| FGR | 69.5 | 0.6 | 117.3 | 1.3 | – | – |
| GEO | 7.43 | 0.19 | 28.38 | 0.69 | – | – |
| ROI (500) | 5.64 | 0.15 | 21.94 | 0.53 | – | – |
| ROI (5000) | 5.44 | 0.15 | 22.17 | 0.53 | – | – |
| AMR | 5.0 | **0.13** | 20.5 | 0.53 | – | – |
| Eda (RK4, 50) | **2.38** | 0.17 | **8.57** | **0.4** | 78.32 | 2.74 |

We report the results in Tab 2. On 3DM and 3DL, we observe that Eda outperforms the baseline methods by a large margin, especially for rotation errors, where Eda achieves more than $50\%$ lower rotation errors on both 3DL and 3DM. We provide more details of Eda on 3DL in Fig. 5 in the appendix.

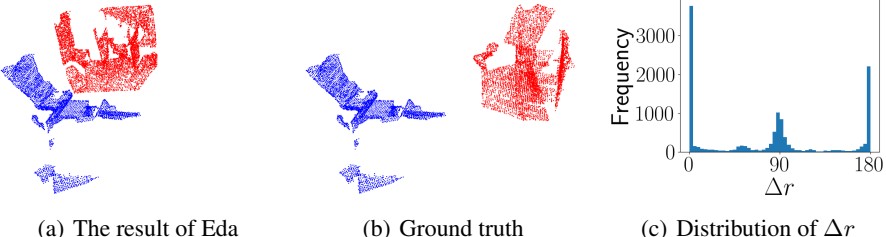

(a) The result of Eda      (b) Ground truth      (c) Distribution of $\Delta r$

Figure 2: More details of Eda on 3DZ. A result of Eda is shown in (a) ($\Delta r = 90.2$). Two PC pieces are marked by different colors. $\Delta r$ is centered at 0, 90, and 180 on the test set (c), suggesting that Eda learns to keeps the orthogonality or parallelism of walls, floors and ceilings of the indoor scenes.

As for 3DZ, we only report the results of Eda in Tab 2, because all baseline methods are not applicable to 3DZ, *i.e.*, their training goal is undefined when the correspondence does not exist. We observe that Eda's error on 3DZ is much larger compared to that on 3DL, suggesting that there exists much larger ambiguity. We provide an example of the result of Eda in Fig. 2. One important observation is that despite the ambiguity of the data, Eda learned the global geometry of the indoor scenes, in the sense that it tends to place large planes, *i.e.*, walls, floors and ceilings, in a parallel or orthogonal position.

To show that this behavior is consistent in the whole test set, we present the distribution of $\Delta r$ of Eda on 3DZ in Fig. 2(c). A simple intuition is that for rooms consisting of 6 parallel or orthogonal planes (four walls, a floor and a ceiling), if the orthogonality or parallelism of planes is correctly maintained in the assembly, then $\Delta r$ should be 0, 90, or 180. We observe that this is indeed the case in Fig. 2(c), where $\Delta r$ is centered at 0, 90, and 180. We remark that the ability to learn global geometric properties beyond correspondences is a key advantage of Eda, and it partially explains the superior performance of Eda in Tab. 2

## 6.3 Multi-piece assembly

This section evaluates Eda on the volume constrained version of BB dataset [35]. We consider the shapes with $2 \leq N \leq 8$ pieces in the "everyday" subset. We compare Eda against the following baseline methods: DGL [49], LEV [43], GLO [35] and JIG [27]. JIG is correspondence-based, and

other baseline methods are regression-based. Note that we do not report the results of the diffusion-type method [34] due to accessibility issues. We process all fragments by grid downsampling with a grid size 0.02 for Eda. For the baseline methods, we follow their original preprocessing steps. To reproduce the results of the baseline methods, we use the implement of DGL and GLO in the official benchmark suite of BB, and we use the official implement of LEV and JIG.

The results are shown in Tab. 3, where we also report the computation time for the whole test set containing 6904 shapes on a Nvidia T4 GPU. We observe that Eda outperforms all baseline methods by a large margin at a moderate computation cost. We present some qualitative results from Fig. 6 to 8 in the appendix, where we observe that Eda can generally reconstruct the shapes more accurately than the baseline methods. An example of the assembly process of Eda is presented in Fig. 3.

Table 3: Quantitative results on BB dataset and the total computation time on the test set.

|  | $\Delta r$ | $\Delta t$ | Time (min) |
|---|---|---|---|
| GLO | 126.3 | 0.3 | **0.9** |
| DGL | 125.8 | 0.3 | **0.9** |
| LEV | 125.9 | 0.3 | 8.1 |
| JIG | 106.5 | 0.24 | 122.2 |
| Eda (RK1, 10) | 80.64 | **0.16** | 19.4 |
| Eda (RK4, 10) | **79.2** | **0.16** | 76.9 |

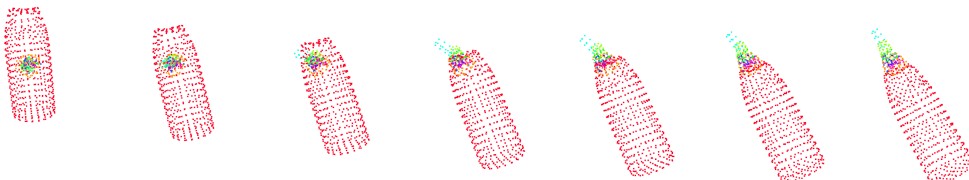

Figure 3: From left to right: the assembly process of a 8-piece bottle by Eda.

### 6.4 Ablation on the number of pieces

This section investigates the influence of the number of pieces on the performance of Eda. We use the kitti odometry dataset [13] containing PCs of city road views. For each sequence of data, we keep pieces that are at least 100 meters apart so that they do not necessarily overlap, and we downsample them using grid downsampling with a grid size 0.5. We train Eda on all consecutive pieces of length $2 \sim N_{max}$ in sequences $0 \sim 8$. We call the trained model Eda-

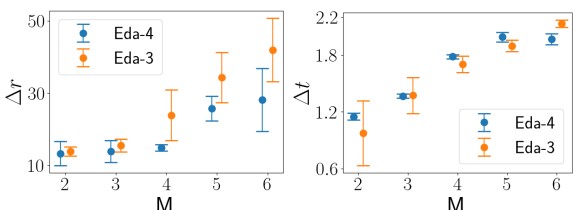

Figure 4: The results of Eda on different number of pieces.

$N_{max}$. We then evaluate Eda-$N_{max}$ on all consecutive pieces of length $M$ in sequence $9 \sim 10$.

The results are shown in Fig. 4. We observe that for $\Delta r$, when the length of the test data is seen in the training set, i.e., $M \leq N_{max}$, Eda performs well, and $M > N_{max}$ leads to worse performance. In addition, Eda-4 generalizes better than Eda-3 on data of unseen length (5 and 6). The result indicates the necessity of using training data of similar length to the test data. Meanwhile, the translation errors of Eda-4 and Eda-3 are comparable, and they both increase with the length of test data.

## 7 Conclusion and discussion

This work studied the theory of equivariant flow matching, and presented a multi-piece assembly method, called Eda, based on the theory. We show that Eda can accurately assemble PCs on practical datasets.

Eda in its current form has several limitations. First, Eda is slow when using a high order RK solver with a large number of steps. Besides its iterative nature, another cause is the lack of kernel level optimization like FlashAttention [8] for equivariant attention layers. We expect to see acceleration in the future when such optimization is available. Second, Eda always uses all input pieces, which is not suitable for applications like archeology reconstruction, where the input data may contain pieces from unrelated objects. Finally, we have not studied the scaling law [19] of Eda in this work, where we expect to see that an increase in model size leads to an increase in performance similar to image generation applications [29].

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

## A More details of the related tasks

The registration task aims to reconstruct the scene from multiple overlapped views. A registration method generally consists of two stages: first, each pair of pieces is aligned using a pair-wise method [30], then all pieces are merged into a complete shape using a synchronization method [1, 21, 15]. In contrast to other tasks, the registration task generally assumes that the pieces are overlapped. In other words, it assumes that some points observed in one piece are also observed in the other piece, and the goal is to match the points observed in both pieces, *i.e.*, corresponding points. The state-of-the-art registration methods usually infer the correspondences based on the feature similarity [47] learned by neural networks, and then align them using the SVD projection [2] or RANSAC.

The robotic manipulation task aims to move one PC to a certain position relative to another PC. For example, one PC can be a cup, and the other PC can be a table, and the goal is to move the cup on the table. Since the input PCs are sampled from different objects, they are generally non-overlapped. Unlike the other two tasks, this task is generally formulated in a probabilistic setting, as the solution is generally not unique. Various probabilistic models, such as energy based model [36, 31], or diffusion model [32], have been used for this task.

The reassembly task aims to reconstruct the complete object from multiple fragment pieces. This task is similar to the registration task, except that the input PCs are sampled from different fragments, thus they are not necessarily overlapped, *e.g.*, due to missing pieces or the erosion of the surfaces. Most of the existing methods are based on regression, where the solution is directly predicted from the input PCs [43, 5, 40]. Some probabilistic methods, such as diffusion based methods [45, 34], have also been proposed. Note that there exist some exceptions [27] which assume the overlap of the pieces, and they reply on the inferred correspondences as the registration methods.

A comparison of these three tasks is presented in Tab. 4.

Table 4: Comparison between registration, reassembly and manipulation tasks.

| Task | Number of pieces | Probabilistic/Deterministic | Overlap |
|---|---|---|---|
| Registration | 2 [30] or more [15] | Deterministic | Overlapped |
| Reassembly | $\geq 2$ | Deterministic | Non-overlapped |
| Manipulation | 2 | Probabilistic | Non-overlapped |
| Assembly (this work) | $\geq 2$ | Probabilistic | Non-overlapped |

## B Connections with bi-equivariance

This section briefly discusses the connections between Def. 3.1 and the equivariances defined in [32] and [40] in pair-wise assembly tasks.

We first recall the definition of the probabilistic bi-equivariance.

**Definition B.1** (Eqn. (10) in [32] and Def. (1) in [33]). $\hat{P} \in \mu(SE(3))$ is bi-equivariant if for all $g_1, g_2 \in SO(3)$, PCs $X_1, X_2$, and measurable set $A \subseteq SE(3)$,

$$\hat{P}(A|X_1, X_2) = \hat{P}(g_2 A g_1^{-1}|g_1 X_1, g_2 X_2). \tag{15}$$

Note that we only consider $g_1, g_2 \in SO(3)$ instead of $g_1, g_2 \in SE(3)$ because we require all input PCs, *i.e.*, $X_i, g_i X_i, i = 1, 2$, to be centered.

Then we recall Def. 3.1 for pair-wise assembly tasks:

**Definition B.2** (Restate $SO(3)^2$-equivariance and $SO(3)$-invariance in Def. 3.1 for pair-wise problems). Let $X_1, X_2$ be the input PCs and $P \in \mu(SE(3) \times SE(3))$.

- $P$ is $SO(3)^2$-equivariant if $P(A|X_1, X_2) = P(A(g_1^{-1}, g_2^{-1})|g_1 X_1, g_2 X_2)$ for all $g_1, g_2 \in SO(3)$ and $A \subseteq SO(3) \times SO(3)$, where $A(g_1^{-1}, g_2^{-1}) = \{(a_1 g_1^{-1}, a_2 g_2^{-1}) : (a_1, a_2) \in A\}$.

- $P$ is $SO(3)$-invariance if $P(A|X_1, X_2) = P(rA|X_1, X_2)$ for all $r \in SO(3)$ and $A \subseteq SO(3) \times SO(3)$.

568 Intuitively, both Def. B.1 and Def. B.2 describe the equivariance property of an assembly solution, and
569 the only difference is that Def. B.1 describes the special case where $X_1$ can be rigidly transformed and
570 $X_2$ is fixed, while Def. B.2 describes the solution where both $X_1$ and $X_2$ can be rigidly transformed.
571 In other words, a solution satisfying Def. B.2 can be converted to a solution satisfying Def. B.1 by
572 fixing $X_2$. Formally, we have the following proposition.

573 **Proposition B.3.** *Let $P$ be $SO(3)^2$-equivariant and $SO(3)$-invariant. If $\tilde{P}(A|X_1, X_2) \triangleq P(A \times$*
574 *$\{e\}|X_1, X_2)$ for $A \subseteq SO(3)$, then $\tilde{P}$ is bi-equivariant.*

575 *Proof.* We prove this proposition by directly verifying the definition.

$$\tilde{P}(g_2 A g_1^{-1}|g_1 X_1, g_2 X_2) = P(g_2 A g_1^{-1} \times \{e\}|g_1 X_1, g_2 X_2) \tag{16}$$
$$= P(g_2 A \times \{e\}|X_1, g_2 X_2) \tag{17}$$
$$= P(A \times \{g_2^{-1}\}|X_1, g_2 X_2) \tag{18}$$
$$= P(A \times \{e\}|X_1, X_2) \tag{19}$$
$$= \tilde{P}(A|X_1, X_2). \tag{20}$$

576 Here, the second and the fourth equation hold because $P$ is $SO(3)^2$-equivariant, the third equation
577 holds because $P$ is $SO(3)$-invariant, and the first and last equation are due to the definition. $\qquad\square$

578 We note that the deterministic definition of bi-equivariance in [40] is a special case of Def. B.1, where
579 $\hat{P}$ is a Dirac delta function. In addition, as discussed in Appx. E in [40], a major limitation of the
580 deterministic definition of bi-equivariance is that it cannot handle symmetric shapes. In contrast, it is
581 straightforward to see that the probabilistic definition, *i.e.*, both Def. B.1 and Def. B.2 are free from
582 this issue. Here, we consider the example in [40]. Assume that $X_1$ is symmetric, *i.e.*, there exists
583 $g_1 \in SO(3)$ such that $g_1 X_1 = X_1$. Under Def. B.1, we have $P(A|X_1, X_2) = P(A|g_1 X_1, X_2) =$
584 $P(A g_1|X_1, X_2)$, which simply means that $P(A|X_1, X_2)$ is $\mathcal{R}_{g_1}$-invariant. Note that this will not
585 cause any contradiction, *i.e.*, the feasible set is not empty. For example, a uniform distribution on
586 $SO(3)$ is $\mathcal{R}_{g_1}$-invariant.

587 As for the permutation-equivariance, the swap-equivariance in [40] is a deterministic pair-wise
588 version of the permutation-equivariance in Def. B.2, and they both mean that the assembled shape is
589 independent of the order of the input pieces.

## C  Proofs

### C.1  Proof in Sec. 4.2

592 To prove Thm. 4.2, which established the relations between related vector fields and equivariant
593 distributions, we proceed in two steps: first, we prove lemma C.1, which connects related vector
594 fields to equivariant mappings; then we prove lemma. C.2, which connects equivariant mappings to
595 equivariant distributions.

596 **Lemma C.1.** *Let $G$ be a smooth manifold, $F : G \to G$ be a diffeomorphism. If vector field $v_\tau$ is*
597 *$F$-related to vector field $w_\tau$ for $\tau \in [0, 1]$, then $F \circ \phi_\tau = \psi_\tau \circ F$, where $\phi_\tau$ and $\psi_\tau$ are generated by*
598 *$v_\tau$ and $w_\tau$ respectively.*

599 *Proof.* Let $\tilde{\psi}_\tau \triangleq F \circ \phi_\tau \circ F^{-1}$. We only need to show that $\tilde{\psi}_\tau$ coincides with $\psi_\tau$.

600 We consider a curve $\tilde{\psi}_\tau(F(\boldsymbol{g}_0))$, $\tau \in [0, 1]$, for a arbitrary $\boldsymbol{g}_0 \in G$. We first verify that $\tilde{\psi}_0(F(\boldsymbol{g}_0)) =$
601 $F \circ \phi_0 \circ F^{-1} \circ F(\boldsymbol{g}_0) = F(\boldsymbol{g}_0)$. Note that the second equation holds because $\phi_0(\boldsymbol{g}_0) = \boldsymbol{g}_0$, *i.e.*, $\phi_\tau$

is an integral path. Then we verify

$$\frac{\partial}{\partial \tau}(\tilde{\psi}_\tau(F(\boldsymbol{g}_0))) = \frac{\partial}{\partial \tau}(F \circ \phi_\tau(\boldsymbol{g}_0)) \tag{21}$$

$$= F_{*,\phi_\tau(\boldsymbol{g}_0)} \circ \frac{\partial}{\partial \tau}(\phi_\tau(\boldsymbol{g}_0)) \tag{22}$$

$$= F_{*,\phi_\tau(\boldsymbol{g}_0)} \circ v_\tau(\phi_\tau(\boldsymbol{g}_0)) \tag{23}$$

$$= w_\tau(F \circ \phi_\tau(\boldsymbol{g}_0)) \tag{24}$$

$$= w_\tau(\tilde{\psi}_\tau(F(\boldsymbol{g}_0))) \tag{25}$$

where the 2-nd equation holds due to the chain rule, and the 4-th equation holds becomes $v_\tau$ is $F$-related to $w_\tau$. Therefore, we can conclude that $\tilde{\psi}_\tau(F(\boldsymbol{g}_0))$ is an integral curve generated by $w_\tau$ starting from $F(\boldsymbol{g}_0)$. However, by definition of $\psi_\tau$, $\psi_\tau(F(\boldsymbol{g}_0))$ is also the integral curve generated by $w_\tau$ and starts from $F(\boldsymbol{g}_0)$. Due to the uniqueness of integral curves, we have $\tilde{\psi}_\tau = \psi_\tau$. □

**Lemma C.2.** *Let $\phi, \psi, F : G \to G$ be three diffeomorphisms satisfying $F \circ \phi = \psi \circ F$. We have $F_\#(\phi_\#\rho) = \psi_\#(F_\#\rho)$ for all distribution $\rho$ on $G$.*

*Proof.* Let $A \subseteq G$ be a measurable set. We first verify that $\phi^{-1}(F^{-1}(A)) = F^{-1}(\psi^{-1}(A))$: If $x \in \phi^{-1}(F^{-1}(A))$, then $(F \circ \phi)(x) \in A$. Since $F \circ \phi = \psi \circ F$, we have $(\psi \circ F)(x) \in A$, which implies $x \in F^{-1}(\psi^{-1}(A))$, *i.e.*, $\phi^{-1}(F^{-1}(A)) \subseteq F^{-1}(\psi^{-1}(A))$. The other side can be verified similarly. Then we have

$$(F_\#(\phi_\#\rho))(A) = \rho(\phi^{-1}(F^{-1}(A))) = \rho(F^{-1}(\psi^{-1}(A))) = (\psi_\#(F_\#\rho))(A), \tag{26}$$

which proves the lemma. □

Now, we can prove Thm. 4.2 using the above two lemmas.

*Proof of Thm. 4.2.* Since $v_X$ is $F$-related to $v_Y$, according to lemma C.1, we have $F \circ \phi_X = \phi_Y \circ F$. Then according to lemma C.2, we have $F_\#(\phi_{X\#}P_0) = \phi_{Y\#}(F_\#P_0)$. The proof is complete by letting $P_X = \phi_{X\#}P_0$ and $P_Y = \phi_{Y\#}(F_\#P_0)$. □

We remark that our theory extends the results in [20], where only invariance is considered, Specifically, we have the following corollary.

**Corollary C.3** (Thm 2 in [20]). *Let $G$ be the Euclidean space, $F$ be a diffeomorphism on $G$, and $v_\tau$ be a $F$-invariant vector field, i.e., $v_\tau$ is $F$-related to $v_\tau$, then we have $F \circ \phi_\tau = \phi_\tau \circ F$, where $\phi_\tau$ is generated by $v_\tau$.*

*Proof.* This is a direct consequence of lemma. C.1 where $G$ is the Euclidean space and $w_\tau = v_\tau$. □

Note that the terminology used in [20] is different from ours: The $F$-invariant vector fields in our work is called $F$-equivariant vector field in [20], and [20] does not consider general related vector fields.

Finally, we present the proof of Prop. 4.5 and Prop. 4.6.

*Proof of Prop. 4.5.* If $v_X$ is $\mathcal{R}_{\boldsymbol{g}^{-1}}$-related to $v_{\boldsymbol{g}X}$, we have $v_{\boldsymbol{g}X}(\hat{\boldsymbol{g}}\boldsymbol{g}^{-1}) = (\mathcal{R}_{\boldsymbol{g}^{-1}})_{*,\hat{\boldsymbol{g}}}v_X(\hat{\boldsymbol{g}})$ for all $\hat{\boldsymbol{g}}, \boldsymbol{g} \in SE(3)^N$. By letting $\boldsymbol{g} = \hat{\boldsymbol{g}}$, we have

$$v_X(\boldsymbol{g}) = (\mathcal{R}_{\boldsymbol{g}})_{*,e}v_{\boldsymbol{g}X}(e) \tag{27}$$

where $(\mathcal{R}_{\boldsymbol{g}})_{*,e} = \left((\mathcal{R}_{\boldsymbol{g}^{-1}})_{*,\boldsymbol{g}}\right)^{-1}$ due to the chain rule of $\mathcal{R}_{\boldsymbol{g}}\mathcal{R}_{\boldsymbol{g}^{-1}} = e$.

On the other hand, if Eqn. (27) holds, we have

$$(\mathcal{R}_{\boldsymbol{g}^{-1}})_{*,\hat{\boldsymbol{g}}}v_X(\hat{\boldsymbol{g}}) = (\mathcal{R}_{\boldsymbol{g}^{-1}})_{*,\hat{\boldsymbol{g}}}(\mathcal{R}_{\hat{\boldsymbol{g}}})_{*,e}v_{\hat{\boldsymbol{g}}X}(e) = (\mathcal{R}_{\hat{\boldsymbol{g}}\boldsymbol{g}^{-1}})_{*,e}v_{\hat{\boldsymbol{g}}X}(e) = v_{\boldsymbol{g}X}(\hat{\boldsymbol{g}}\boldsymbol{g}^{-1}), \tag{28}$$

which suggests that $v_X$ is $\mathcal{R}_{\boldsymbol{g}^{-1}}$-related to $v_{\boldsymbol{g}X}$. Note that the second equation holds due to the chain rule of $\mathcal{R}_{\boldsymbol{g}^{-1}}\mathcal{R}_{\hat{\boldsymbol{g}}} = \mathcal{R}_{\hat{\boldsymbol{g}}\boldsymbol{g}^{-1}}$, and the first and the third equation are the result of Eqn. (27). □

*Proof of Prop. 4.6.* 1) Assume $v_X$ is $\sigma$-related to $v_{\sigma X}$: $(\sigma)_{*,g} v_X(g) = V_{\sigma X}(\sigma(g))$. By inserting Eqn. (5) to this equation, we have

$$(\sigma)_{*,\boldsymbol{g}}(\mathcal{R}_{\boldsymbol{g}})_{*,e} f(\boldsymbol{g}X) = (\mathcal{R}_{\sigma\boldsymbol{g}})_{*,e} f(\sigma(\boldsymbol{g})\sigma(X)). \tag{29}$$

Since $\sigma \circ \mathcal{R}_{\boldsymbol{g}} = \mathcal{R}_{\sigma\boldsymbol{g}} \circ \sigma$, by the chain rule, we have $\sigma_*(\mathcal{R}_{\boldsymbol{g}})_* = (\mathcal{R}_{\sigma\boldsymbol{g}})_* \sigma_*$. In addition, $\sigma(\boldsymbol{g})\sigma(X) = \sigma(\boldsymbol{g}X)$. Thus, this equation can be simplified as

$$(\mathcal{R}_{\sigma\boldsymbol{g}})_* \sigma_* f(\boldsymbol{g}X) = (\mathcal{R}_{\sigma\boldsymbol{g}})_{*,e} f(\sigma(\boldsymbol{g}X)) \tag{30}$$

which suggests

$$\sigma_* f = f \circ \sigma. \tag{31}$$

The first statement in Prop. 4.6 can be proved by reversing the discussion.

2) Assume $v_X$ is $\mathcal{L}_r$-related to $v_X$: $(\mathcal{L}_r)_{*,g} v_X(\boldsymbol{g}) = V_X(r\boldsymbol{g})$. By inserting Eqn. (5) to this equation, we have

$$(\mathcal{L}_r)_{*,\boldsymbol{g}}(\mathcal{R}_{\boldsymbol{g}})_{*,e} f(\boldsymbol{g}X) = (\mathcal{R}_{r\boldsymbol{g}})_{*,e} f(r\boldsymbol{g}X). \tag{32}$$

Since $\mathcal{R}_{r\boldsymbol{g}} = \mathcal{R}_{\boldsymbol{g}} \circ \mathcal{R}_r$, by the chain rule, we have $(\mathcal{R}_{r\boldsymbol{g}})_{*,e} = (\mathcal{R}_{\boldsymbol{g}})_{*,r}(\mathcal{R}_r)_{*,e}$. In addition, $(\mathcal{L}_r)(\mathcal{R}_{\boldsymbol{g}}) = (\mathcal{R}_{\boldsymbol{g}})(\mathcal{L}_r)$, by the chain rule, we have $(\mathcal{L}_r)_{*,\boldsymbol{g}}(\mathcal{R}_{\boldsymbol{g}})_{*,e} = (\mathcal{R}_{\boldsymbol{g}})_{*,r}(\mathcal{L}_r)_{*,e}$. Thus the above equation can be simplified as

$$(\mathcal{L}_r)_{*,e} f(\boldsymbol{g}X) = (\mathcal{R}_r)_{*,e} f(r\boldsymbol{g}X) \tag{33}$$

which implies

$$f \circ r = (\mathcal{R}_{r^{-1}})_{*,r} \circ (\mathcal{L}_r)_{*,e} \circ f. \tag{34}$$

By representing $f$ in the matrix form, we have

$$w_\times^i(rX) = r w_\times^i(X) r^T \tag{35}$$

$$t^i(rX) = r t^i(X) \tag{36}$$

for all $i$, where $r$ on the right hand side represents the matrix form of the rotation $r$. Here the first equation can be equivalently written as $w^i(rX) = r w^i(X)$. The second statement in Prop. 4.6 can be proved by reversing the discussion. $\qquad\square$

## C.2 Proofs in Sec. 4.3

To establish the results in this section, we need to assume the uniqueness of $r^*$ (6):

**Assumption C.4.** The solution to (6) is unique.

Note that this assumption is mild. A sufficient condition [40] of assumption C.4 is that the singular values of $\tilde{\boldsymbol{g}}_1^T \boldsymbol{g}_0 \in \mathbb{R}^{3\times3}$ satisfy $\sigma_1 \geq \sigma_2 > \sigma_3 \geq 0$, *i.e.*, $\sigma_2$ and $\sigma_3$ are not equal. We leave the more general treatment without requiring the uniqueness of $r^*$ to future work.

We first justify the definition of $\boldsymbol{g}_1 = r^* \tilde{\boldsymbol{g}}_1$ by showing that $\boldsymbol{g}_1$ follows $P_1$ in the following proposition.

**Proposition C.5.** *Let $P_0$ and $P_1$ be two $SO(3)$-invariant distributions, and $\boldsymbol{g}_0$, $\tilde{\boldsymbol{g}}_1$ be independent samples from $P_0$ and $P_1$ respectively. If $r^*$ is given by (6) and assumption C.4 holds, then $\boldsymbol{g}_1 = r^* \tilde{\boldsymbol{g}}_1$ follows $P_1$.*

*Proof.* Define $A_{\tilde{\boldsymbol{g}}_1} = \{\boldsymbol{g}_0 | r^*(\boldsymbol{g}_0, \tilde{\boldsymbol{g}}_1) = e\}$, where we write $r^*$ as a function of $\tilde{\boldsymbol{g}}_1$ and $\boldsymbol{g}_0$. Then we have $P(r^* = e|\tilde{\boldsymbol{g}}_1) = P_0(A_{\tilde{\boldsymbol{g}}_1})$ by definition. In addition, due to the uniqueness of the solution to (6), for an arbitrary $\hat{r} \in SO(3)$, we have $P(r^* = \hat{r}|\tilde{\boldsymbol{g}}_1) = P_0(\hat{r}A_{\tilde{\boldsymbol{g}}_1})$. Since $P_0$ is $SO(3)$-invariant, we have $P_0(\hat{r}A_{\tilde{\boldsymbol{g}}_1}) = P_0(A_{\tilde{\boldsymbol{g}}_1})$, thus, $P(r^* = \hat{r}|\tilde{\boldsymbol{g}}_1) = P(r^* = e|\tilde{\boldsymbol{g}}_1)$. In other words, for a given $\tilde{\boldsymbol{g}}_1$, $r^*$ follows the uniform distribution $U_{SO(3)}$.

Finally we compute the probability density of $\boldsymbol{g}_1$:

$$P(\boldsymbol{g}_1) = \int P(r^* = \hat{r}^{-1}|\hat{r}\boldsymbol{g}_1) P_1(\hat{r}\boldsymbol{g}_1) d\hat{r} \tag{37}$$

$$= \int U_{SO(3)}(\hat{r}) P_1(\boldsymbol{g}_1) d\hat{r} \tag{38}$$

$$= P_1(\boldsymbol{g}_1), \tag{39}$$

which suggests that $\boldsymbol{g}_1$ follows $P_1$. Here the second equation holds because $P_1$ is $SO(3)$-invariant. $\qquad\square$

668 Then we discuss the equivariance of the constructed $h_X$ (7).

669 **Proposition C.6.** *Given $r \in SO(3)^N$, $g_0, \tilde{g}_1 \in SE(3)^N$, $\sigma \in S_N$, $r \in SO(3)$ and $\tau \in [0, 1]$. Let*
670 *$h_X$ be a path generated by $g_0$ and $\tilde{g}_1$. Under assumption C.4,*

671 • *if $h_{rX}$ is generated by $g_0 r^{-1}$ and $\tilde{g}_1 r^{-1}$, then $h_{rX}(\tau) = \mathcal{R}_{r^{-1}} h_X(\tau)$.*

672 • *if $h_{\sigma X}$ is generated by $\sigma(g_0)$ and $\sigma(\tilde{g}_1)$, then $h_{\sigma X}(\tau) = \sigma(h_X(\tau))$.*

673 • *if $\hat{h}_X$ is generated by $r g_0$ and $r \tilde{g}_1$, then $\hat{h}_X(\tau) = \mathcal{L}_r(h_X(\tau))$.*

674 *Proof.* 1) Due to the uniqueness of the solution to (6), we have $r^*(g_0 r^{-1}, \tilde{g}_1 r^{-1}) = r^*(g_0, \tilde{g}_1)$.
675 Thus, we have

$$h_{rX}(\tau) = \exp(\tau \log(g_1 g_0^{-1})) g_0 r^{-1} = \mathcal{R}_{r^{-1}}(h_{rX}(\tau)). \tag{40}$$

676 2) Due to the uniqueness of the solution to (6), we have $r^*(\sigma(g_0), \sigma(\tilde{g}_1)) = \sigma(r^*(g_0, \tilde{g}_1))$. Thus,
677 we have $\sigma(h_X) = h_{\sigma X}$.

678 3) Due to the uniqueness of the solution to (6), we have $r^*(r g_0, r \tilde{g}_1) = r r^*(g_0, \tilde{g}_1) r^{-1}$. Thus,

$$\hat{h}_{rX}(\tau) = \exp(\tau \log(r r^* \tilde{g}_1 g_0^{-1} r^{-1})) r g_0 = r \exp(\tau \log(r^* \tilde{g}_1 g_0^{-1})) g_0 = \mathcal{L}_r(h_X(\tau)). \tag{41}$$

679 □

680 With the above preparation, we can finally prove Prop. 4.7.

681 *Proof of Prop. 4.7.* 1) By definition

$$L(rX) = \mathbb{E}_{\tau, g_0' \sim P_0, \tilde{g}_1' \sim P_{rX}} ||v_{rX}(h_{rX}(\tau)) - \frac{\partial}{\partial \tau} h_{rX}(\tau)||_F^2, \tag{42}$$

682 where $h_{rX}$ is the path generated by $g_0'$ and $\tilde{g}_1'$. Since $P_0 = (\mathcal{R}_{r^{-1}})_\# P_0$ and $P_{rX} = (\mathcal{R}_{r^{-1}})_\# P_X$ by
683 assumption, we can write $g_0' = g_0 r^{-1}$ and $\tilde{g}_1' = \tilde{g}_1 r^{-1}$, where $g_0 \sim P_0$ and $\tilde{g}_1 \sim P_X$. According to
684 the first part of Prop. C.6, we have $h_{rX}(\tau) = \mathcal{R}_{r^{-1}} h_X(\tau)$, where $h_X$ is a path generated by $g_0$ and $\tilde{g}_1$.
685 By taking derivative on both sides of the equation, we have $\frac{\partial}{\partial \tau} h_{rX}(\tau) = (\mathcal{R}_{r^{-1}})_{*, h_X(\tau)} \frac{\partial}{\partial \tau} h_X(\tau)$.
686 Then we have

$$L(rX) = \mathbb{E}_{\tau, g_0' \sim P_0, \tilde{g}_1' \sim P_{rX}} ||v_{rX}(\mathcal{R}_{r^{-1}} h_X(\tau)) - (\mathcal{R}_{r^{-1}})_{*, h_X(\tau)} \frac{\partial}{\partial \tau} h_X(\tau)||_F^2 \tag{43}$$

687 by inserting these two equations into Eqn. (42). Since $v_X$ is $\mathcal{R}_{r^{-1}}$-related to $v_{rX}$ by assumption, we
688 have $v_{rX}(\mathcal{R}_{r^{-1}} h_X(\tau)) = (\mathcal{R}_{r^{-1}})_{*, h_X(\tau)} v_X(h_X(\tau))$. Thus, we have

$$||v_{rX}(\mathcal{R}_{r^{-1}} h_X(\tau)) - (\mathcal{R}_{r^{-1}})_{*, h_X(\tau)} \frac{\partial}{\partial \tau} h_X(\tau)||_F^2 = ||(\mathcal{R}_{r^{-1}})_{*, h_X(\tau)} (v_{rX}(h_X(\tau)) - \frac{\partial}{\partial \tau} h_X(\tau))||_F^2$$

$$= ||(v_{rX}(h_X(\tau)) - \frac{\partial}{\partial \tau} h_X(\tau))||_F^2 \tag{44}$$

689 where the second equation holds because $(\mathcal{R}_{r^{-1}})_{*, h_X(\tau)}$ is an orthogonal matrix. The desired result
690 follows.

691 2) The second statement can be proved similarly as the first one, where $\sigma$-equivariance is considered
692 instead of $\mathcal{R}_{r^{-1}}$-equivariance.

693 3) Denote $g_0' = r g_0$ and $\tilde{g}_1' = r \tilde{g}_1$, where $g_0 \sim P_0$ and $\tilde{g}_1 \sim P_X$. According to the third part of
694 Prop. C.6, we have $\hat{h}_X(\tau) = \mathcal{L}_r(h_X(\tau))$. By taking derivative on both sides of the equation, we
695 have $\frac{\partial}{\partial \tau} \hat{h}_X(\tau) = (\mathcal{L}_r)_{*, h_X(\tau)} \frac{\partial}{\partial \tau} h_X(\tau)$. Then the rest of the proof can be conducted similarly to the
696 first part of the proof. □

697 # D $SO(2)$-reduction

698 The main idea of $SO(2)$-reduction [] is to rotate the edge $uv$ to the $y$-axis, and then update node
699 feature in the rotated space. Since all 3D rotations are reduced to 2D rotations about the $y$-axis in the
700 rotated space, the feature update rule is greatly simplified.

Here, we describe this technique in the matrix form to facilitates better parallelization. The original element form description can be found in []. Let $F_v^l \in \mathbb{R}^{c \times (2l+1)}$ be a $c$-channel $l$-degree feature of point $v$, and $L > 0$ be the maximum degree of features. We construct $\hat{F}_v^l \in \mathbb{R}^{c \times (2L+1)}$ by padding $F_v^l$ with $L - l$ zeros at the beginning and the end of the feature, then we define the full feature $F_v \in \mathbb{R}^{c \times L \times (2L+1)}$ as the concatenate of all $\hat{F}_v^l$ with $0 < l \leq L$. For an edge $vu$, there exists a rotation $r_{vu}$ that aligns $uv$ to the $y$-axis. We define $R_{vu} \in \mathbb{R}^{L \times (2L+1) \times (2L+1)}$ to be the full rotation matrix, where the $l$-th slice $R_{vu}[l, :, :]$ is the $l$-th Wigner-D matrix of $r_{vu}$ with zeros padded at the boundary. $K_v$ defined in (11) can be efficiently computed as

$$K_v = R_{vu}^T \times_{1,2} (W_K \times_3 (D_K \times_{1,2} R_{vu} \times_{1,2} F_v)), \tag{45}$$

where $M_1 \times_i M_2$ represents the batch-wise multiplication of $M_1$ and $M_2$ with the $i$-th dimension of $M_2$ treated as the batch dimension. $W_K \in \mathbb{R}^{(cL) \times (cL)}$ is a learnable weight, $D_K \in \mathbb{R}^{c \times (2L+1) \times (2L+1)}$ is a learnable matrix taking the form of 2D rotations about the $y$-axis, *i.e.*, for each $i$, $D_K[i, :, :]$ is

$$\begin{bmatrix} a_1 & & & & & & & & & -b_1 \\ & a_2 & & & & & & & -b_2 & \\ & & \ddots & & & & & \cdot^{\cdot^{\cdot}} & & \\ & & & a_{L-1} & & -b_{L-1} & & & & \\ & & & & a_L & & & & & \\ & & & b_{L-1} & & a_{L-1} & & & & \\ & & \cdot^{\cdot^{\cdot}} & & & & \ddots & & & \\ & b_2 & & & & & & a_2 & & \\ b_1 & & & & & & & & & a_1 \end{bmatrix}, \tag{46}$$

where $a_1, \cdots, a_L, b_1, \cdots, b_{L-1} : \mathbb{R}_+ \to \mathbb{R}$ are learnable functions that map $|vu|$ to the coefficients. $V_v$ defined in (11) can be computed similarly. Note that (45) does not require the computation of Clebsch-Gordan coefficients, the spherical harmonic functions, and all computations are in the matrix form where no for-loop is needed, so it is much faster than the computations in (11).

# E  More details of Sec. 6

We present more details of Eda on 3DL in Fig. 5. We observe that the vector field is is gradually learned during training, *i.e.*, the training error converges. On the test set, RK4 outperforms the RK1, and they both benefit from more time steps, especially for rotation errors.

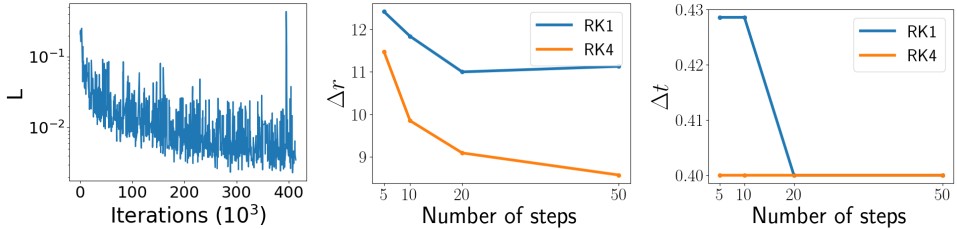

Figure 5: More details of Eda on 3DL. Left: the training curve. Middle and right: the influence of RK4/RK1 and the number of time steps on $\Delta r$ and $\Delta t$.

We provide the complete version of Table 2 in Table 5, where we additionally report the standard deviations of Eda.

We provide some qualitative results on BB datasets in Fig. 6 and Fig. 8. Eda can generally recover the shape of the objects except for some rare cases, such as the $3rd$ sample in the second row in Fig. 6. We hypothesize that Eda can achieve better performance when using finer grained inputs. A complete version of Tab. 3 is provided in Tab. 6, where we additionally report the standard deviations of Eda.

We provide a few examples of the reconstructed road views in Fig. 9.

Table 5: The complete version of Table 2 with stds of Eda reported in bracked.

| | 3DM | | 3DL | | 3DZ | |
|---|---|---|---|---|---|---|
| | $\Delta r$ | $\Delta t$ | $\Delta r$ | $\Delta t$ | $\Delta r$ | $\Delta t$ |
| FGR | 69.5 | 0.6 | 117.3 | 1.3 | – | – |
| GEO | 7.43 | 0.19 | 28.38 | 0.69 | – | – |
| ROI (500) | 5.64 | 0.15 | 21.94 | 0.53 | – | – |
| ROI (5000) | 5.44 | 0.15 | 22.17 | 0.53 | – | – |
| AMR | 5.0 | **0.13** | 20.5 | 0.53 | – | – |
| Eda (RK4, 50) | **2.38** (0.16) | 0.16 (0.01) | **8.57** (0.08) | **0.4** (0.0) | 78.74 (0.6) | 0.96 (0.01) |

Table 6: The complete version of Table 3 with stds of Eda reported in brackets.

| | $\Delta r$ | $\Delta t$ | Time (min) |
|---|---|---|---|
| GLO | 126.3 | 0.3 | 0.9 |
| DGL | 125.8 | 0.3 | 0.9 |
| LEV | 125.9 | 0.3 | 8.1 |
| Eda (RK1, 10) | 80.64 | 0.16 | 19.4 |
| Eda (RK4, 10) | 79.2 (0.58) | 0.16 (0.0) | 76.9 |

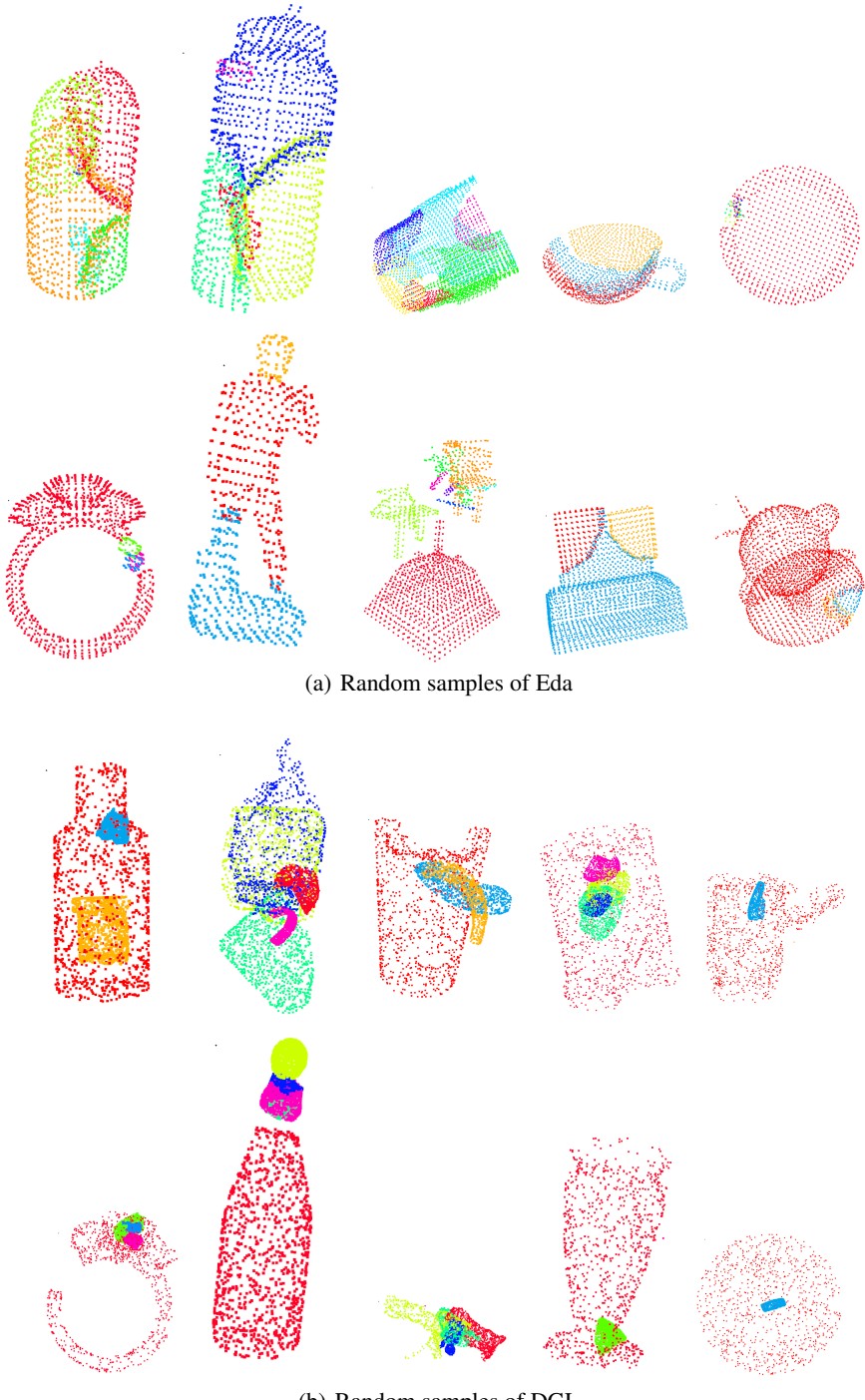

(a) Random samples of Eda

(b) Random samples of DGL

Figure 6: Qualitative results of Eda and DGL.

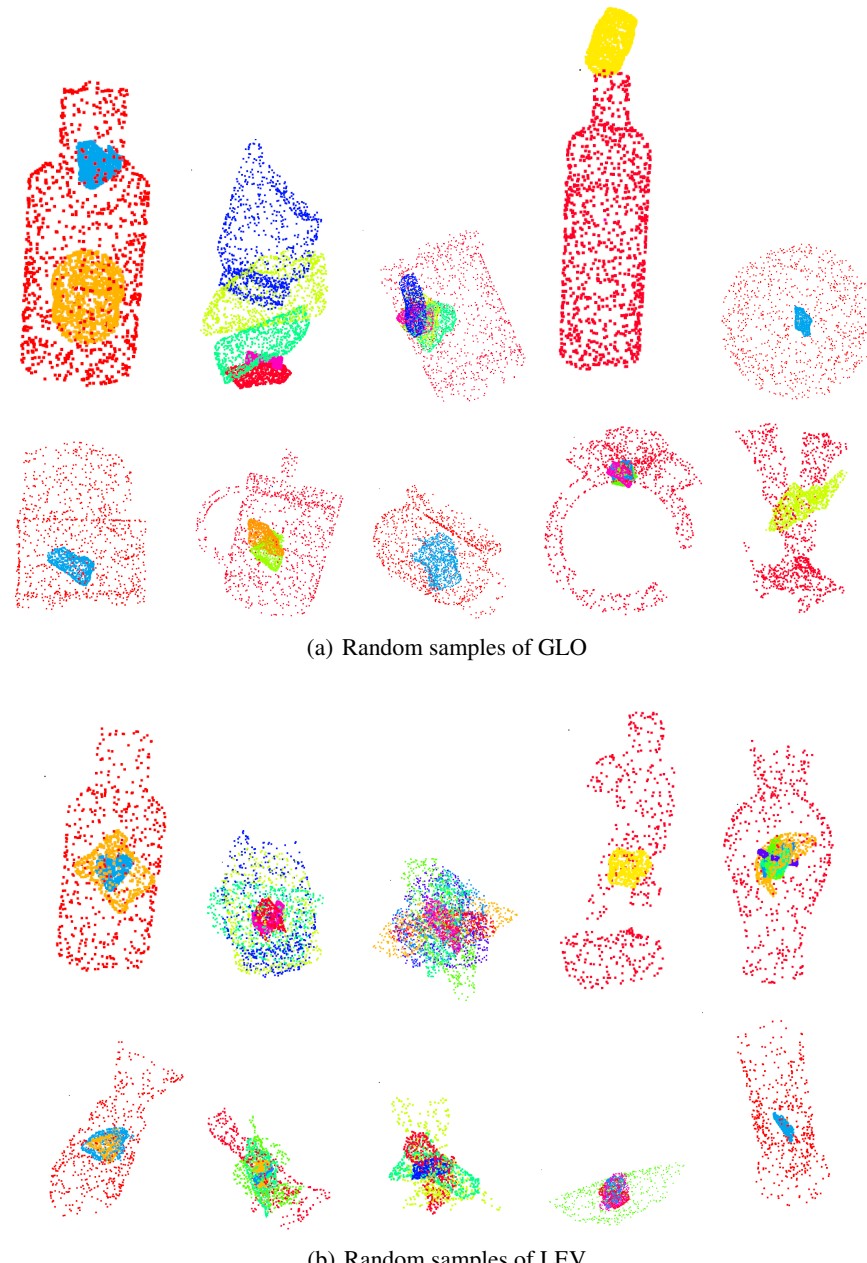

(a) Random samples of GLO

(b) Random samples of LEV

Figure 7: Qualitative results of GLO and LEV.

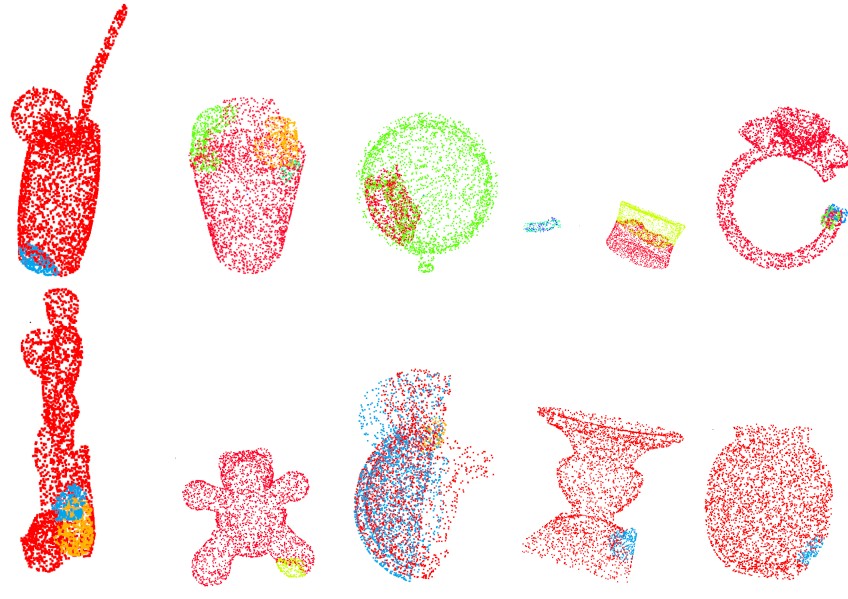

(a) Random samples of JIG

Figure 8: Qualitative results of JIG.

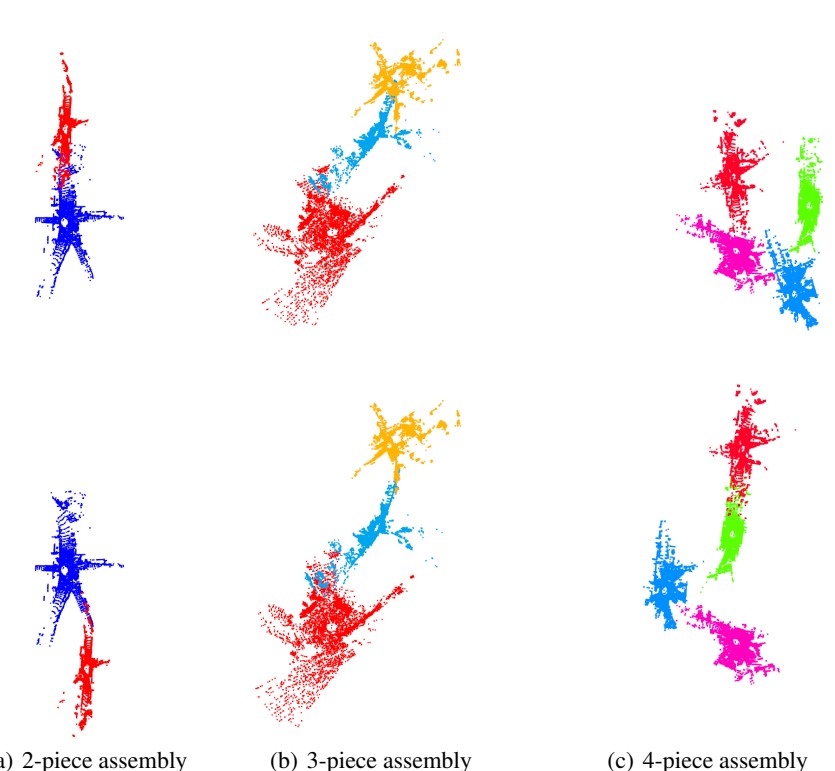

(a) 2-piece assembly    (b) 3-piece assembly    (c) 4-piece assembly

Figure 9: Qualitative results of Eda on kitti. We present the results of Eda (1-st row) and the ground truth (2-nd row). For each assembly, Eda correctly places the input road views on the same plane.

