# OpenReview forum: "Equivariant Flow Matching for Point Cloud Assembly"
_NeurIPS.cc/2025/Conference — Submitted to NeurIPS 2025_

### Official Review · Reviewer_yNZU · 2025-06-24

**Clarity:** 2
**Significance:** 3
**Originality:** 3
**Rating:** 3
**Confidence:** 3

**Summary:**

The submitted work tackles the task of point-cloud assembly and specifically focuses on assembly of multi-piece objects (the method will use all pieces to reconstruct the full shape, i.e. its not possible to omit pieces). To this end, authors propose an equivariant flow-matching method. The main finding is that learning an equivariant distribution via flow-matching can be solved by finding related vector fields (which later need to be integrated using e.g. Runge-Kutta method to obtain the point-cloud assembly from the predicted vector field).  This makes point cloud assmebly via equivariant flow-matching computationally tractable. The proposed method shows improved results according to reported metrics.

**Questions:**

- l. 345: the qualitative comparison between methods seems impossible since you chose different objects to assemble for the different methods in Fig 6 and 8 in the appendix
- l.373-375: the concluding sentence about kernel level optimization is unlcear and should be rewritten to improve clarity.
- is there a specific reason why the scaling law was not studied? if not it should be added.
- l. 698 citation is empty
- fig 5. right: why is there such a large increase in training error in the one of the last training iterations? (even higher than the initial error)

**Ethical Concerns:**

["NO or VERY MINOR ethics concerns only"]

**Final Justification:**

While i appreciate the authors efforts to improve the paper, i think it needs further revision to improve readability and evaluation (qualitative comparison). As i consider these changes to require substantial modifications to the paper which cannot be reviewed appropriately, I vote for rejection.

**Limitations:**

yes

**Quality:**

3

**Strengths And Weaknesses:**

## Strengths
- the paper is well motivated and (apart from mathematical details) mostly well written
- the paper provides strong technical contributions which is backed up with mathematical derivations
- the method shows promising results on multiple datasets in multiple settings according to reported metrics

## Weaknesses
- while the paper provides solid technical contributions, these contributions are hidden under a very cluttered and heavy notation. Furthermore, no intuitive figures are provided which could help the reader understand key concepts more easily.
- evaluation metrics which are used in the experiments are not explained and with that evaluation is not fully clear. In addition, qualitative comparison to other methods is not possible (see questions section).

## Minor Weaknesses
- the rendering of point clouds are hard to interpret as lack of shadows make it hard to perceive their 3D geometry
- proofs in the appendix should be referenced from the main paper, e.g. the proof of Theorem 4.2 should be referenced after Theorem 4.2.

---

> ### Author Rebuttal · Authors · 2025-07-29
>
> We thank the reviewer for the time and effort. We address the concerns below.
>
>
> 1. **while the paper provides solid technical contributions, these contributions are hidden under a very cluttered and heavy notation. Furthermore, no intuitive figures are provided which could help the reader understand key concepts more easily.**
>
> To improve the readability, we have now added a complete walk-through of a toy example in the appendix (Please see the "more intuition" part in our reply to reviewer i6up). We have also provided brief readable comments for Prop.4.5 and 4.6 (See our reply to reviewer ia2b quesion 2 for details) We hope those improvements can make the reading easier.
>
>
>
> 2. **evaluation metrics which are used in the experiments are not explained and with that evaluation is not fully clear.**
>
> The evaluation metric is the pair-wise rotation/translation error. For each pair of piece $(i, j)$, we compute the error of $i$ w.r.t. $j$. We have now added the following clarification in Sec. 6.1: **This metric is the pair-wise rotation/translation error: it measures the averaged error of $g_i$ w.r.t $g_j$ for all $(i, j)$ pairs of pieces.**
>
>
>
>
> 3. **the rendering of point clouds are hard to interpret as lack of shadows make it hard to perceive their 3D geometry**
>
> We apologize for the lack of clarity. We used Open3D for visualization, however its rendering ability is limited.
>
> To provide better visualizations, we have now added more visualizations of Eda on 3DZ in Fig 10 in the appendix, where the camera is set to look at the room from above, so the main structures of the rooms are clearer. (Note that we are not allowed to upload figures during this rebuttal.)
>
>
>
>
> 5. **proofs in the appendix should be referenced from the main paper, e.g. the proof of Theorem 4.2 should be referenced after Theorem 4.2.**
>
> Thanks for the suggestion. We have now added a sentence: **All proofs can be found in Appx.C** in the introduction section.
>
>
> Q1. **l. 345: the qualitative comparison between methods seems impossible since you chose different objects to assemble for the different methods in Fig 6 and 8 in the appendix**
>
> To avoid cherry picking, we used random samples in each class instead of fixed samples. Nevertheless, there are some overlaps of sample selection, for example, the ring shape has appeared in Fig. 6(a),  6(b), 7(a) and 8, where a direct comparison is possible.
>
>
> Q2. **l.373-375: the concluding sentence about kernel level optimization is unlcear and should be rewritten to improve clarity.**
>
> We apologize for the confusion. The "kernel" means CUDA kernel of the Nvidia GPU [8]. We have now made this clear: **...CUDA kernel level...**
>
>
> Q3. **is there a specific reason why the scaling law was not studied? if not it should be added.**
>
> The focus of this work is a new theory and its algorithm. Since we have experimentally shown that the algorithm is effective on existing datasets, we think it is appropriate to leave the study of the scaling law to future work. We have now modified the last sentence to make this clearer: **Finally, the scaling law [19] of Eda is an interesting research direction left for future work...**
>
>
> Q4. **l. 698 citation is empty**
>
> Thanks for pointing this out. We have now fixed that citation ([28]).
>
> Q5. **fig 5. right: why is there such a large increase in training error in the one of the last training iterations? (even higher than the initial error)**
>
> This is a spike in the training loss. Such spikes are commonly seen in the loss curve of transformer-type models. See [1] for an in-depth discussion.
>
> [1] Takase, Sho, et al. "Spike no more: Stabilizing the pre-training of large language models." arXiv preprint arXiv:2312.16903 (2023).

---

> > ### Comment · Reviewer_yNZU · 2025-08-04
> >
> > Thank you for your reply and clarifications.
> >
> > 1. Thank you for providing more intuition for the problem. I nevertheless still believe that especially the notation could be simplified by removing excessive amounts of sub or superscripts
> >
> > 5. I think it would be more convenient for the reader to reference the proofs in the appendix at the respective text sections.
> >
> > Q1. I appreciate that the goal is to avoid cherry-picking but it would have been possible to sample random pairs and report these random pairs for all methods. Even though the results contain some overlap, imo the qualitative comparison still is not useful without a true side-to-side comparison (as mentioned above).
> >
> > All other points have been sufficiently discussed, thank you.

---

> ### Author Response · Authors · 2025-08-04
>
> Thanks for your reply!
> 1. Thanks for your suggestion. We have now removed the (*) symbol in the subscript in Prop 4.5 and the paragraph after it, as the expression would be computationally (not mathematically) equivalent. However, we feel it is necessary to keep other subscripts, like $X$,  (#), and $i$, as they represent the conditions, the pushforward, and the index of point cloud respectively. Removing them will cause some confusions.
> 2. We have now added a reference to the proof after each proposition.
> 3. Thanks for your suggestion. We will select more results, and place them side-by-side for better comparison.

---

> > ### Comment · Reviewer_yNZU · 2025-08-05
> >
> > Thank you. I have no further questions.

---

### Official Review · Reviewer_ia2b · 2025-07-01

**Clarity:** 2
**Significance:** 3
**Originality:** 3
**Rating:** 3
**Confidence:** 3

**Summary:**

The paper tackles point cloud assembly. It tries to prove in theory that learning equivariant distributions via flow matching is to learn related vector fields. Corresponding method is proposed based on the theory, called equivariant diffusion assembly. Experiments are done on 3DMatch and BB.

**Questions:**

See weaknesses. Frankly speaking, I cannot fully understand the theory, especially the meaning of Proposition 4.5 and 4.6. I understand most of the training process and experiments, so my rating is based on these parts. My main concern is the ablation study can not support the theory. If the author can help me understand the value of the theory and solve my concerns in limited experiments, I will recommend to accept this paper.

**Ethical Concerns:**

["NO or VERY MINOR ethics concerns only"]

**Final Justification:**

The authors clarfies the motivation and the meaning of the theory in the rebuttal, but I am not sure if all these explanations will be included in the revised version due to the page limitations. The writing needs significant improvement. Also, I think the experiments are insufficient. Although I think this paper conveys valuable insight to the community, I cannot give a positive recommendation based on this manuscript. However, I believe this paper would have a decent chance to get accepted with enhanced readability.

**Limitations:**

yes

**Quality:**

2

**Strengths And Weaknesses:**

Strengths:
1. Performance is good.
2. Theory seems to be interesting, but I cannot not fully undrestand.

Weaknesses:
1. Insufficient ablation study: The paper does not sufficiently validate the individual contributions of its components. This significantly weakens the experimental section. For example, the distribution of noise; the choice of equivariant and permutation-invariant network f; optimization in SE(3) space; rotation correction in Eq 6.
2. Clarity of writing: I worked on the point cloud assembly before and I think many readers has similar background as mine. The theory part is really difficult to understand. The definition of the symbol is unclear (L121-125) and some background knowledge should be introduced in a more detailed way before moving on. Proposition 4.5 and 4.6 seems to be very important, maybe a brief comment using natural language will help the readers to understand the paper. Also I think the bottom-right corner of se3 matrix in Eq 5 should be 1.
3. Experiments in zero-overlap setting: It seems that the rotation error is very large (78.32) and it is said that the distribution of error is centered at 0, 90, 180. The meaning of this experiment is unclear. Although I agree that "Eda learns to keeps the orthogonality or parallelism of walls, floors and ceilings of the indoor scenes", it seems that all it learns is this global geometric constraints instead of part assembly. Moreover, the visualization in Figure 2 is hard to understand.

---

> ### Author Rebuttal · Authors · 2025-07-29
>
> We thank the reviewer for the time and effort. We address the concerns below.
>
>
> 1. **Insufficient ablation study: The paper does not sufficiently validate the individual contributions of its components. This significantly weakens the experimental section. For example, the distribution of noise; the choice of equivariant and permutation-invariant network f; optimization in SE(3) space; rotation correction in Eq 6.**
>
> Thanks for your suggestion. As for ablation study, we did an ablation on the Rounge-Kutta degree (1/4) and step (from $5$ to $50$) in Fig. 5 in the appendix. We have now added an ablation study of the rotation correction. We compare Eda with and without rotation correction on 3DL and 3DM in the following table. We observe that the Eda without rotation correction performs worse than Eda, while it still performs better than all baselines in Tab.2. We have now added this table to the appendix.
>
> |   |  3DM | 3DM  | 3DL   | 3DL  |
> |---|---|---|---|---|
> |   |  $\Delta r$  |  $\Delta t$ | $\Delta r$  | $\Delta t$ |
> | Eda  | 2.4  |0.16  | 8.5| 0.4 |
> |  Eda w/o R  | 3.4   |  0.2 |13.5 | 0.6 |
>
>
>
>
> As for other components, we do not feel necessary to discuss the use of them because we never claim any contribution on these components, and they are alreay sufficiently simple or are standard.
>
> - The noise $(U_{SO(3)} \otimes \mathcal{N})^N$ is the simplest noise we can think of: the rotation component is a uniform distribution, which does not contain any preference; its translation component is a Gaussian noise, which is standard for Euclidean space, and they are independently coupled. We do not know any simpler alternative to this noise.
> - The network is designed to be nothing more than an "imitation of the standard transformer structure" as stated in line 227. We combined Croco [42],  Equiformer[23] and DiT (diffusion transformer) [29], which are already standard.
> - The optimiztion is in $se(3)$ (a Lie algebra), which is a linear space. The training loss is the standard mean square error loss (Eqn 2) in flow matching.
>
>
>
> 2.  **The definition of the symbol is unclear (L121-125) and some background knowledge should be introduced in a more detailed way before moving on. Proposition 4.5 and 4.6 seems to be very important, maybe a brief comment using natural language will help the readers to understand the paper.**
>
>
> Due to the limited space, we can not present more preliminaries on related vector fields, so we cited an introductional text book [37], where the reader can get a quick idea of the concepts. We have now made this clearer: **More details can be found in Sec.14.6 in [37].**
>
> Intuitively, the "relatedness" of two vector fields $v_{X}$ and $v_{gX}$ means that $v_{X}$ is a "transformed" version of $v_{gX}$. Corollary 4.4 first suggests that to keep the $SO(3)^N$-equivariance of solution, $v_{X}$ must be a transformed version of $v_{gX}$. Then, **Proposition 4.5** suggests that to satisfy such requirement of $v_X$ and $v_{gX}$, equation $v_X(g)=v_{gX}(e)g$ must hold. This is useful because it provides a way to represent $v_{X}$ as a neural network, i.e., in Eqn (4), when we parameterize $v_{gX}(e)$ on the right hand side as $f(gX)$, then $v_X(g)=f(gX)g$. Futhermore, **Proposition 4.6** translates the other two requirements of $v_{X}$ (permutation and $SO(3)$-relatedness) to the requirements of $f$.
>
> The comments of these results were already in line 163-165 and line 170-172. Following your suggestion, we have now rewritten them for readability. Line 163: **Prop. 4.5 provides a way to represent $v_X$ by a neural network. Specifically,...**. Line 170: **To guarantee $\sigma$-relatedness and  $\mathcal{L}_r$-invariance of $v_X$, the following requirements of $f$ are needed.**. We have further made proposition 4.6 clearer by replacing the relatedness notation $\\#$ by natural language: in (1) **...., then $v_X$ is $\sigma$-related to  $v_{\sigma X}$**. and in (2) **..., then $v_X$ is $\mathcal{L}_r$-invariant**.
>
> Please also refer to the added "more intuition" section in our reply to reviewer i6up, where we provide a walk-through of a toy example. A concrete formulation of Prop.4.5 and 4.6 is also provided there.
>
>
> 3.  **Also I think the bottom-right corner of se3 matrix in Eq 5 should be 1.**
>
>
> No. It is $0$. a $se(3)$ matrix (a Lie algebra) is not a $SE(3)$ (a Lie group) matrix.  $se(3)$ can be seen as a gradient of $SE(3)$. The bottom-right corner of $SE(3)$ is a constant $1$, so after taking a gradient, that element is $0$ in $se(3)$.
>
>
>
> 4. **Experiments in zero-overlap setting: It seems that the rotation error is very large (78.32) and it is said that the distribution of error is centered at 0, 90, 180. The meaning of this experiment is unclear.**
>
> The meaning of the experiments is to show that Eda can "learn global geometric properties beyond correspondences"  even when the true assembly is ambiguous due to the lack of overlap as noted in line 328.  This is "a key advantage of Eda" compared to all baseline methods in Tab.2,  which all rely on correspondence, i.e.,   they cannot learn such geometry properties in this experiment.
>
> The statistics of rotation error was reported to show that the model learned plausible solutions instead of doing random guessing despite a large mean error, i.e. if the model is doing random guessing, then the rotation error distribution would be close to a uniform distribution on interval [0, 180].
>
>
> 5. **Although I agree that "Eda learns to keeps the orthogonality or parallelism of walls, floors and ceilings of the indoor scenes", it seems that all it learns is this global geometric constraints instead of part assembly.  Moreover, the visualization in Figure 2 is hard to understand.**
>
> Being able to keep the orthogonality/parallelism of planes already shows that Eda can learn the rotation component of the assembly.
> In addition,
> as shown in the visualization in Fig.2(a),
> Eda can also learn the translation faithfully, i.e., it keeps a plausible distance between walls of the assembled room, while keeping the ceilings (floors) on the same plane.
>
>
> As for the visualiztion, we apologize for the lack of clarity. We have now added more visualizations of Eda on 3DZ in Fig 10 in the appendix for better understanding,
> where the camera is set to look at the room from above.
> (Note that we are not allowed to upload figures during this rebuttal.)

---

> > ### Comment · Reviewer_ia2b · 2025-08-05
> >
> > Thanks for the response. The clarification is clear. However, I still think some ablation study should be essential to prove your theory. The current experiments cannot fully prove that your theory contributes to the performance gain. For example, use a network $f$ which is not permutation-equivariant or SO(3)-equivariant.

---

> ### Author Response · Authors · 2025-08-06
>
> Thanks for your reply.
>
> We believe that the experiments are sufficient to show the usefulness of the theory becaseu our model is built solely on the theory, and it is shown to be SOTA. We do not think more ablation studies are neede because we have never claimed the performance gain of using any individual part of the theory, like the networks, or the mean square error training.
>
> In particular, the advantages of the SO(3)-equivariant networks are proved to be the ensurance of the SO(3)-invariance of the solution, but we have never claimed (or proved) the performance gain of using those networks, so we do not see why an ablation on the networks is needed: replacing the equivariant network by a non-equivariant one may lead to better, worse, or comparable performance, and none of these results conflicts with our theory.
>
> If the reviewer wants to see the contribution of equivariances on performance, a recent diffusion-based model AMR in Tab. 2 could be compared, which is permutation-equivariant but not SO(3)-equivariant.

---

> > ### Comment · Reviewer_ia2b · 2025-08-06
> >
> > Thanks for your response. I have no further questions.

---

### Official Review · Reviewer_xTSh · 2025-07-02

**Clarity:** 3
**Significance:** 3
**Originality:** 3
**Rating:** 5
**Confidence:** 3

**Summary:**

This work proposes an equivariant flow matching framework for multi-piece point cloud assembly tasks. The key idea is to employ a vector field parameterized by equivariant networks on an invariant base distribution to ensure the output distribution is equivariant to SO(3) rotations and permutation. Additionally, the training efficiency is enhanced by considering modified samples and random noises with minimum distance across all possible rotations. Overall, the experimental results show that the proposed framework achieves better results than existing baselines.

**Questions:**

In the third condition of Corollary 4.4, it seems that equivariance of the vector field to SO(3) is already sufficient to ensure the invariance of the generation distribution (as suggested in [1]). Considering that invariance is a stricter requirement, I wonder if it is necessary?

**Ethical Concerns:**

["NO or VERY MINOR ethics concerns only"]

**Final Justification:**

The rebuttal has addressed my concerns, and I will maintain the rating.

**Limitations:**

The number of pieces is predetermined in the current framework. It would be valuable to discuss the possibility of extending the framework to handle a variable number of pieces.

**Paper Formatting Concerns:**

Nil

**Quality:**

3

**Strengths And Weaknesses:**

Strength:
- This work proposes a clear formulation of the base distribution and vector field network assumptions to ensure the output distribution is equivariant to the group. The formulation of the base distribution $\left(U_{\mathrm{SO}(3)} \otimes \mathcal{N}(0, \omega I)\right)^N$ and equivariant layers appear to be well-suited for these assumptions.
- The experimental results demonstrate strong performance of the proposed framework, achieving better results in both pair-wise registration and multi-piece assembly.

Weakness:
There are some minor concerns:
- The idea of parameterizing vector fields using equivariant networks has been explored in other domains, e.g., 3D module generation [1,2]. Despite the differences in applications, the network architectures and training strategy (Optimal Transport Training) seem to be highly related. It would be beneficial to cite and discuss these works.
- Since the proposed framework is generative in nature, it would be reasonable to investigate the diversity of the generated solutions. In particular, the variance of the evaluation metrics should be reported, and multiple solutions could be visualized given the same set of inputs, especially for cases without significant overlaps.

[1] Equivariant flow matching. NeurIPS 2023.

[2] Equivariant Flow Matching with Hybrid Probability Transport for 3D Molecule Generation. NeurIPS 2023.

I recommend accepting this paper due to its strong theoretical foundation in equivariant flow matching for point cloud assembly tasks. The authors make important contributions by (1) establishing a theoretical framework that reduces equivariant distribution learning to the simpler task of constructing related vector fields, (2) developing a novel assembly model (Eda) with guaranteed equivariance properties, and (3) demonstrating superior performance on challenging datasets including non-overlapped pieces. While there are minor concerns about citing related work and reporting solution diversity, the paper's technical novelty and strong empirical results justify acceptance.

---

> ### Author Rebuttal · Authors · 2025-07-29
>
> We thank the reviewer for the positive feedback. We now address the concerns below.
>
> 1. **The idea of parameterizing vector fields using equivariant networks has been explored in other domains, e.g., 3D module generation [1,2]. Despite the differences in applications, the network architectures and training strategy (Optimal Transport Training) seem to be highly related. It would be beneficial to cite and discuss these works.**
>
> Thanks for your suggestion. Those two papers are indeed related to our work.
>
> Compared to [1,2], a key difference of our work in theory is that it compares different "equivariant" or "related" vector fields, e.g, $v_X$ and $v_{gX}$, instead of one "invariant" vector field (the word "invariant" in our work means "equivariant" in [1,2]. Aslo see our reply 3 for this difference.). This is noted in line 65. On the other hand, [1, 2] developed an OT training strategy for flow matching on invariant distributions which is beyond the discussion of our work. Nevertheless, we use the rotation correction (6) which is similar to the first equation in (16) in [1], and we believe that the use of the second equation in (16), i.e., OT, is useful for future applications.
>
> We have now added the following sentence in line 66 in the related work section: **Furthermore, when the distribution is invariant, the optimal-transport-based integral path [53, 54] was explored for efficient sampling.**
> and we have cited them in line 196: **A similar rotation correction in the Euclidean space was studied in [53, 54].**
>
> 2. **...In particular, the variance of the evaluation metrics should be reported, and multiple solutions could be visualized given the same set of inputs, especially for cases without significant overlaps.**
>
> Thanks for the suggestion. The variance was already reported in Tab. 5 and Tab. 6 in the appendix.
>
> We have now added a figure showing 3 different runs of Eda on 6 samples of 3DZ in Fig. 10 in the appendix. (Note that we are not allowed to upload figures during this rebuttal.)
>
>
> 3. **In the third condition of Corollary 4.4, it seems that equivariance of the vector field to SO(3) is already sufficient to ensure the invariance of the generation distribution (as suggested in [1]). Considering that invariance is a stricter requirement, I wonder if it is necessary?**
>
> Yes. It is necessary.
> This is because the "equivariance" of vector field in [1,2] is called "invariance" in our work,
> so Corollary 4.4 in our work and Theorem 4.1 in [2] use the same concept.
> (We save the word "equivariance" for comparing two different vector fields.)
> We noted such difference in line 624.
>
>
> 4. **The number of pieces is predetermined in the current framework. It would be valuable to discuss the possibility of extending the framework to handle a variable number of pieces.**
>
> No. The number of pieces can vary. This is because the $SO(3)^N$-equivariance is ensured by construction: the network computes the velocity $(w, t)$ for each of the $N$ pieces, where the network (transformer structure) does not assume a fixed length $N$. For example, in our experiments, the BB dataset contains $2 \sim  8$ pieces, and the kitti dataset contains $2 \sim M$ pieces, where $M$ is $3$ or $4$.

---

> ### Author Response · Authors · 2025-08-06
>
> Dear reviewer xTSh, please let us know whether we have addressed your questions and concerns.

---

> > ### Comment · Reviewer_xTSh · 2025-08-06
> >
> > The rebuttal has addressed my concerns, and I maintain my positive rating on the submission.

---

### Official Review · Reviewer_i6up · 2025-07-03

**Clarity:** 1
**Significance:** 2
**Originality:** 2
**Rating:** 3
**Confidence:** 2

**Summary:**

In this work, the proposed model, Eda, is an equivariant method for multi-piece assembly based on flow matching.
Eda uses invariant noise and predicts related vector fields by construction, allowing faster processing; it does not involve handling the SE(3)^N -equivariance for the N-piece assembly task. Eda is evaluated on rotated 3Dmatch and BB dataset, along with ablation on the number of pieces in the assembly task.

**Questions:**

Q1. Why is there a need to compute the rotation error (equation 6) if the path is SO(3) equivariant?

Q2. If the distribution is equivariant, how are the parts in the assembly aligned?

Q3. Proposition 4.7 does not seem clear. Could you elaborate on this?

Q4. In the experimental setup, the center-of-mass frame is considered. Now, suppose all the objects are aligned, i.e., there is a central axis across which the object is present (which is more common for ShapeNet, etc.). How does the Eda use this to be computationally efficient?

Q5. Could you provide intuition for the necessity of using training data of similar length to the test data?

**Ethical Concerns:**

["NO or VERY MINOR ethics concerns only"]

**Final Justification:**

Eda, is an equivariant method for multi-piece assembly based on flow matching. Eda uses invariant noise and predicts related vector fields by construction, allowing faster processing.
My main concerns for the paper were a lack of clarity in writing as well as the method itself, limited ablation, and a lack of intuition for the method. Lack of intuition itself is not a negative, but it was unclear why certain choices were made, which made it very difficult to read the paper. The authors gave some intuition for the method in the rebuttal, and thus, I increased my score.

I do not fully believe this paper is at NeurIPS standard as it needs improved clarity in writing (I agree with reviewer ia2b here). I cannot gauge the importance or relevance of the tasks presented, as it is not my area of expertise.

**Limitations:**

Yes.

**Paper Formatting Concerns:**

No major paper formatting concerns.

**Quality:**

2

**Strengths And Weaknesses:**

### Strengths
- The paper tackles a relevant problem of 3D point cloud assembly by learning equivariant distributions.

### Weakness
- The paper is a bit difficult to read and could benefit from modifying the text (esp Sec 4.2). (There are a few clarification questions below)
- The model evaluation, both wr.t to datasets as well as ablation, is limited.
- The intuition of the proposed method is not clear, and part of the method seems to lack clarity.

---

> ### Author Rebuttal · Authors · 2025-07-29
>
> We thank the reviewer for the time and effort.  We first present the added section to provide more intuitions, and then respond to each of the specific concern.
>
>
> ## More intuition
>
> We have now added the following paragraphs to the appendix to provide some intuitive explanations of the main results without the tools for describing transformations of probabilities (pushforward) and vector fields (relatedness).
> We have now added a sentence in the introduction: **A walk-through of the theory using a toy example is provided in Appx. B.**
>
> ### Sec B. A toy example
> Consider the following two-piece deterministic example.
> Assume that a solution for the input point clouds $(X, Y)$ is $(r_1, r_2)$,
> meaning $r_1X$ and $r_2Y$ are assembled,
> where $r$ is a rotation matrix.
>
> The equivariances in Def 3.1 are natural properties of the solution:
> when $(X, Y)$ are transformed, the solution will change accordingly:
>
>
> 1. $SO(3)^2$-equivariance: a solution for $(r_3X, r_4Y)$ is $(r_1r_3^{-1}, r_2r_4^{-1})$.  Because $(r_1r_3^{-1}r_3X, r_2 r_4^{-1}r_4Y)=(r_1X, r_2Y)$ are assembled by assumption.
> 2. Permutation-equivariance:  a solution for $(Y, X)$ is $(r_2, r_1)$.  Because $(r_2Y, r_1X)$ are assembled by assumption.
> 3. SO(3)-invariance: another solution for $(X, Y)$ is $(rr_1, rr_2)$.  Because $(rr_1X, rr_2Y)$ are just the assembled point clouds $(r_1X, r_2Y)$ rotated by $r$.
>
>
> **Corollary 4.4** incorporates these equivariances into flow matching.
> Denote $v_{(X, Y)}$ the vector field learned for $(X, Y)$.
> For $SO(3)^2$-equivariance,
> the goal is to ensure $v_{(r_3X, r_4Y)}$ flows to $(r_1r_3^{-1}, r_2r_4^{-1})$ when $v_{(X, Y)}$  flows to $(r_1, r_2)$.
> This corollary shows that the goal can be achieved if
> $v_{(r_3X, r_4Y)}$ is a proper "transformation" of $v_{(X, Y)}$ (related),
> and the noise is invariant.
>
>
> **Proposition 4.5** provides a way to construct $v_{(X, Y)}$ satisfying the $SO(3)^2$-equivariance requirement of Corollary 4.4: $v_{(X, Y)}(r_7, r_8) =f(r_7X, r_8Y)(r_7 \oplus r_8)$, here
> $
> f(X, Y) = (w_1, t_1) \oplus(w_2, t_2)
> $
> is a neural network mapping $(X, Y)$ to their respective rotation/translation velocity component $w$ and $t$, and $\oplus$ is the concatenation. (Vector $w$ and $t$ are combined into a matrix as in Eqn. 5.)
>
>
> **Proposition 4.6** suggests that, to ensure the other two requirements (permutation and SO(3)-relatedness) of $v_X$, $f$ needs to satisfy
> $$
> f(Y, X) = (w_2, t_2) \oplus (w_1, t_1) \quad and \quad f(rX, rY) = (rw_1, rt_1)\oplus (rw_2, rt_2)
> $$
>
>
>
> **Proposition 4.7** suggests that some data augmentations are not needed. Specifically, for data $(X, Y)$ we learn a vector field $v_{(X, Y)}$.  We can randomly augment the data $(r_3X, r_4Y)$ and learn $v_{(r_3X, r_4Y)}$.  However, this proposition suggests that this is not necessary when the path and $v_{(X, Y)}$ are "$SO(3)^2$-equivariance" (relatedness) and the noise is invariant.   Similar results hold for the other two types of augmentations.
>
>
> -------
>
> We now address each of the concerns below.
>
>
> 1. **The paper is a bit difficult to read and could benefit from modifying the text (esp Sec 4.2).
>             3. The intuition of the proposed method is not clear, and part of the method seems to lack clarity.
>            Q2.  If the distribution is equivariant, how are the parts in the assembly aligned?
>            Q3.  Proposition 4.7 does not seem clear. Could you elaborate on this?**
>
> For more intuition,
> we have now added a complete walk-through of a toy example in the appendix (see above).
> In particular,
> the meaning of 3 equivariances and the meaning of Prop. 4.7 is included.
> For clarification,
> we have now provided brief and readable comments for Prop.4.5 and 4.6 (A brief comment of Prop. 4.7 are in line 208.) Please see our reply to reviewer ia2b quesion 2 for details.
> We hope the above modifications make the reading easier.
>
> 2. **The model evaluation, both wr.t to datasets as well as ablation, is limited.**
>
> We have included all standard assembly datasets in our experiments.
> i.e.,  3DM, 3DL and kitti.
> See a recent paper AMR[6] for example.
> We additionally considered BB and a novel dataset 3DZ.
> If the reviewer thinks some datasets are missing, please let us know.
>
> We did an ablation study on the Rounge-Kutta degree (1/4) and step (from $5$ to $50$) in Fig. 5 in the appendix.
> We have added an ablation study for rotation correction.
> Please see our reply to reviewer ia2b question 1 for details.
>
>
>
>
>
>
> Q1. **Why is there a need to compute the rotation error (equation 6) if the path is SO(3) equivariant?**
>
> As we stated in line 197,
> the purpose is to remove the "redundant rotation component".
> For an oversimplified example,
> for noise $g_0=(r_1, r_2)$ and a data point $\tilde{g_1}=(rr_1, rr_2)$,
> we can get a clean $g_1=(r_1, r_2)$ where $r$ is removed by solving (6).
> This process is compatible with the path selection.
>
>
> Q4. **In the experimental setup, the center-of-mass frame is considered. Now, suppose all the objects are aligned, i.e., there is a central axis across which the object is present (which is more common for ShapeNet, etc.). How does the Eda use this to be computationally efficient?**
>
>
> Eda is designed to be $SO(3)$-invariant, meaning that it has no preference of canonical poses (like those in ShapeNet).
> To use those canonical poses,
> one needs to remove the $SO(3)$-invariance, which would remove Proposition 4.6 (2),
> meaning equivariant networks are not needed. This might lead to more efficient computation.
>
>
> Q5. **Could you provide intuition for the necessity of using training data of similar length to the test data?**
>
>
> Apology for the confusion of this sentence.
> We mean the test data lengths should be included in training data.
> We correct the sentence in line 367 as **...necessity of using training data whose lengths subsume that of the test data.**
>
> Such requirement of the training data is common for transformer models.
> See Table 2 in [1] for an example.
>
> [1] Sun, Yutao, et al. "A length-extrapolatable transformer." arXiv preprint arXiv:2212.10554 (2022).

---

> > ### Comment · Reviewer_i6up · 2025-08-05
> >
> > I thank the authors for responding to my questions. I have no further questions.

---

### Decision · Program_Chairs · 2025-09-17

**Decision:**

Reject

**Comment:**

This paper addresses the problem of multi-piece point cloud assembly and introduces Eda which is an equivariant flow-matching framework. The key insight is that learning equivariant distributions via flow matching can be achieved by predicting related vector fields from an invariant base distribution, making the task more efficient and computationally tractable. Experiments on benchmark datasets demonstrate consistent improvements over existing baselines.

While one reviewer expressed a positive opinion, the other three reviewers raised significant concerns regarding the clarity and readability of the paper, which maks it difficult to follow the method and its underlying motivations. They also pointed to insufficient ablation studies and limited experimental evaluation, particularly the absence of qualitative comparisons. Although the rebuttal provided additional intuition and clarification, reviewers remained uncertain whether these improvements would be incorporated into the final version given space constraints.

Overall, all three reviewers are on an agreement that the paper requires substantial revisions to improve clarity and strengthen the evaluation before it can be considered for acceptance. The AC agrees with the reviewers and recommends rejection.